# Benchmarking and Enhancing Rational Preference Utilization for Personalized Assistants: A Pragmatic View

## Abstract

Large language model (LLM)-powered assistants have recently integrated memory mechanisms that record user preferences, leading to more personalized and user-aligned responses. However, the dual effects of personalization remain underexplored, and its adverse consequences are especially salient in real-world applications. To address this gap, we propose Rational Personalization Acts, which reformulates memory utilization as a problem of pragmatic intent reasoning. Building on this perspective, we develop RPEVAL, a benchmark comprising a personalized intent reasoning dataset and a multi-granularity evaluation protocol. RPEVAL not only reveals the widespread phenomenon of irrational personalization in existing LLMs, but also, through a novel error pattern analysis, illustrates how irrational personalization can undermine user experience. Finally, we introduce RP-REASONER, which treats memory utilization as a pragmatic reasoning process, enabling the selective integration of personalized information. Experimental results demonstrate that our method significantly outperforms carefully designed baselines on RPEVAL, and resolves 80% of the bad cases observed in a large-scale commercial personalized assistant, highlighting the potential of pragmatic reasoning to mitigate irrational personalization. Our benchmark is publicly available at https://anonymous.4open.science/r/RPEval-E4B0.

## 1 Introduction

**Background.** In human communication, people tend to express themselves as economically as possible—using minimal language to convey maximal intent. This often results in utterances that are heavily underspecified, relying on the listener to fill in the gaps through shared knowledge and mutual understanding (Grice, 1975). This fundamental cognitive mechanism underlies the increasing appeal of personalized assistants (PAs) (Zhang et al., 2024a; Zhao et al., 2025) powered by large language models (LLMs) (OpenAI, 2023). By continuously interacting with users, PAs incrementally build personalized memory stores that capture user-specific information. When handling to new queries, they read relevant memories and incorporate them into the context, enabling the generation of personalized response (Zhang et al., 2024c), thereby enhancing user experience and satisfaction.

**Motivation.** This work centers on the duality of personalization, particularly the potential risks. As shown in Figure 1, a user who usually prefers strong rhythm music might still be recommended fast-paced tracks based on historical preferences when requesting sleep-aid audio—clearly misaligned with the current need. This issue is particularly pronounced in real-world scenarios: (1) **memories are often sparse and fragmented**, leading to oversimplified or overly labeled modeling of users (He et al., 2017); (2) **user queries are typically open-ended**, frequently involving requests unrelated to previously stored information (Chaney et al., 2018). As a result, when generating personalized responses, PAs may inevitably introduce irrelevant memories into the context. This can lead the LLMs to misuse those memories, producing outputs that deviate from the user's actual intent. This observation motivates the roadmap of our work: Define rational personalization, benchmark LLMs against this standard, and explore how LLMs can rationally utilize memory for user-aligned responses.

**Key Insight.** In this work, our central claim is that the true value of memory lies in assisting PAs in better interpreting the user's intent (Zhang et al., 2018; Moon et al., 2019), rather than merely "re-

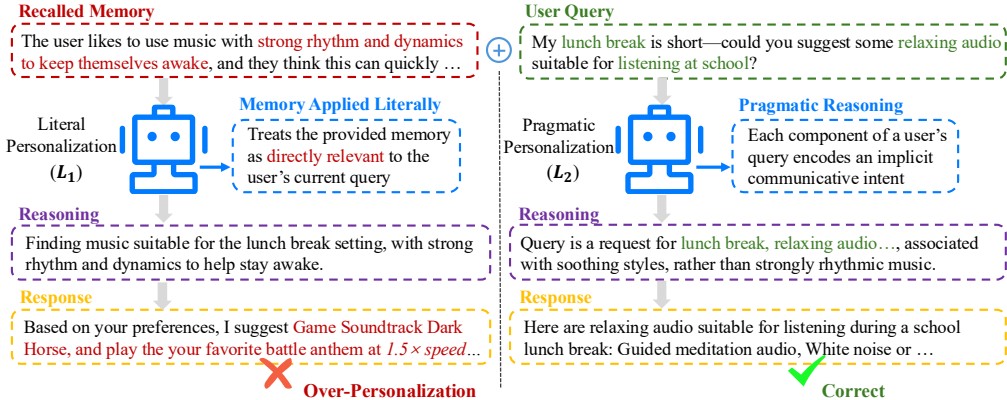

Figure 1: Different levels of PAs. In $L_1$, memory is directly concatenated with the query, whereas in $L_2$, the PA infers implicit cues from the user's query to determine memory utilization strategy.

playing the past." We rethink the role of personalized memory through the lens of pragmatic intent inference, and propose a foundational framework called **R**ational **P**ersonalization **A**cts (**RPA**). The idea is inspired by the Rational Speech Acts theory (Frank & Goodman, 2012), which formalizes how humans infer the underlying intent behind a query through different levels of pragmatic reasoning. As shown in Figure 1, to reveal the core challenges of existing PAs, we divide them into two levels: Most existing PAs remain at the stage of *literal personalization*, where memory is directly concatenated with the context and mechanically incorporated into responses, often leading to **over-personalization**. In contrast, *pragmatic personalization* extracts subtle cues from user queries, interprets these hints to uncover underlying needs, and thereby enables **rational personalization**.

**Benchmark.** Based on RPA, we propose **RPEval**—a benchmark for evaluating the rational personalization capabilities of PAs. RPEval focuses on key challenging scenarios in the real world: when users issue underspecified everyday queries and memories with varying applicability are introduced into a PA's context, the PA must decide whether and how to integrate these memories to accurately infer the user's intent. RPEval consists of two core components. (1) *Personalized Intent Reasoning Dataset*—simulating how users with different preferences express their intentions through natural, underspecified queries. The dataset is constructed under the design principles of *diversity, naturalness, and consistency* via a tailored *bootstrapping–inversion–validation–expansion* pipeline, covering over 8,000 preferences across twelve task categories, providing a solid foundation for evaluation of rational personalization. (2) *Multi-granularity Evaluation Protocol*— Alongside intent classification, RPEval introduces a distinctive *error-pattern–driven* evaluation paradigm. This paradigm not only reveals an accuracy gap of about 40%–90% between mainstream LLMs and human annotators in memory utilization strategies, but also systematically characterizes failure modes arising from irrational personalization, such as `filter bubbles` and `redundant information`.

**Method.** To tackle this challenge, we introduce **RP-Reasoner**, which reformulates personalized memory utilization as a reasoning mechanism grounded in pragmatics. Instead of mechanically concatenating memories, RP-REASONER approximates the user's hidden process of query formulation, extract cues from surface-level language, infer the underlying intent, and selectively integrate personalized information. Experimental results show that RP-REASONER not only improves intent prediction accuracy by about 35%, reduces error severity by 26% across mainstream LLMs on RPEVAL, but also resolves nearly 80% of the bad cases observed in large-scale commercial PAs, highlighting the substantial potential of pragmatic reasoning to enhance the rationality of PAs.

**Contribution.** Our key contributions are as follows:

- We propose a novel pragmatic perspective on memory utilization (RPA), which, to the best of our knowledge, is the first study that systematically analyzes the dual role of personalization in PAs.
- We develop RPEVAL, a benchmark with a personalized intent reasoning dataset and multi-granularity evaluation, which reveals widespread over-personalization in existing LLMs and demonstrates, via novel error analysis, how irrational personalization undermines user experience.
- We introduce RP-REASONER, validated on RPEVAL and a large-scale commercial PAs, highlighting the significant potential of pragmatic reasoning to mitigate irrational personalization.

## 2 RATIONAL PERSONALIZATION ACTS

Drawing on RSA (Frank & Goodman, 2012), we propose Rational Personalization Acts (**RPA**), which categorizes memory utilization strategies of PAs at different levels through the lens of pragmatic intent reasoning, thereby enabling a systematic analysis of the dual effects of personalization.

**Problem Formulation.** In personalized response generation, LLM takes $(m, q)$ as input, where $q$ is the user's current query and $m$ is the dialogue-context memory comprising multiple user preferences $\{p_0, p_1, \ldots, p_K\}$. LLM's goal is to predict the user's intent $i$ and generate a response $r$:

$$(i, r) = \text{LLM}(m, q), \qquad \textit{Evaluation target: } \text{assess the alignment of } i \text{ with } i_{\text{query}}.$$

Here, $i_{\text{query}}$ denotes the true intent underlying $q$. While $m$ may enrich $q$ and support more accurate inference of $i_{\text{query}}$, it can also be irrelevant or even conflict with it. Unlike existing benchmarks (Zhao et al., 2025; Li et al., 2025b; Tan et al., 2025b), we emphasize that the evaluation of $r$ should not be based solely on its consistency with $m$, but rather on its consistency with the user's true intent $i_{\text{query}}$.

**Rational Personalization Acts.** We categorize memory utilization strategies into three rational levels, situating current PAs within a developmental spectrum and revealing their core limitations.

**Definition 1** (Non-personalized Assistant $L_0$). $L_0$ ignores personalized memory and predicts the user's intent solely based on the semantic match between the query and possible intents:

$$P_{L_0}(i, r \mid q) \propto \text{Semantic}(q, i) \cdot P(i),$$

where $\text{Semantic}(q, i)$ measures the surface-level semantic compatibility between $q$ and $i$, and $P(i)$ is the prior probability of intent $i$. Without the support of user memory $m$, $L_0$ tends to produce generic responses, resulting in **under-personalization**. `Example:` When $q$ is "Can you recommend some music?", it can only give a generic recommendation without tailoring to specific style preferences.

**Definition 2** (Literal Personalized Assistant $L_1$). $L_1$ directly appends personalized memory $p$ to the input context and uses it during generation, aiming to produce responses more aligned with $m$:

$$P_{L_1}(i, r \mid q, m) \propto \text{Semantic}(q, m, i) \cdot P(i \mid m),$$

When $m$ is unrelated to the $i_{\text{query}}$, $L_1$ may cause **over-personalization**. `Example:` When $m$ is "The user likes strong rhythm" and $q$ is "Recommend some audio for lunch break?", $L_1$ may still lean toward recommending upbeat music, overlooking the crucial requirement of aiding sleep.

**Definition 3** (Pragmatic Personalized Assistant $L_2$). $L_2$ is guided by the principle that *each component of a user's query encodes an implicit communicative intent*, from which it derives cues for whether and how to integrate personalized memory, thereby enabling more accurate intent inference.

$$P_{L_2}(i, r \mid m, q) \propto P_{\text{user}}(q \mid i, m) \cdot P(i \mid m),$$

Here, $P_{\text{user}}(q \mid i, m)$ denotes the likelihood that a user with profile $m$ would issue query $q$ given intent $i$. $L_2$ approximates the *query formulation process*, enabling reverse inference of latent intent and guiding the selection of the most **rational personalization** strategy. `Example:` When the query $q$ explicitly includes "lunch break, relaxing," $L_2$ reasons that the preference for *strong rhythm* is not suitable in the current context, thereby avoiding being misled by irrelevant memory.

RPA provides a principled framework for PAs from a pragmatic perspective, highlighting both the importance of personalization and its core challenges. Building on this foundation, we introduce RPEVAL, a benchmark for personalized intent understanding that systematically evaluates the rational personalization abilities of LLMs, with empirical results showing that most existing LLMs remain at the $L_1$ stage (§ 3). Finally, we propose RP-REASONER, a reasoning framework for LLMs inspired by pragmatic inference, paving the way toward building pragmatically rational PAs (§ 4).

## 3 RPEVAL: ARE CURRENT LLMS RATIONAL PERSONALIZED ASSISTANTS?

In this section, we introduce RPEVAL, a novel benchmark that casts personalized response generation as a personalized intent reasoning problem. It addresses one of the most challenging real-world settings: when users issue short, underspecified everyday-life queries and the dialogue context introduces memories of varying applicability. PAs must decide whether and how to rationally integrate these memories to recover the true intent. To support this evaluation, RPEVAL consists of two components: a Personalized Intent Reasoning Dataset (§ 3.1) and a Multi-Granularity Evaluation Protocol (§ 3.2), providing a systematic and in-depth evaluation of state-of-the-art LLMs (§ 3.3).

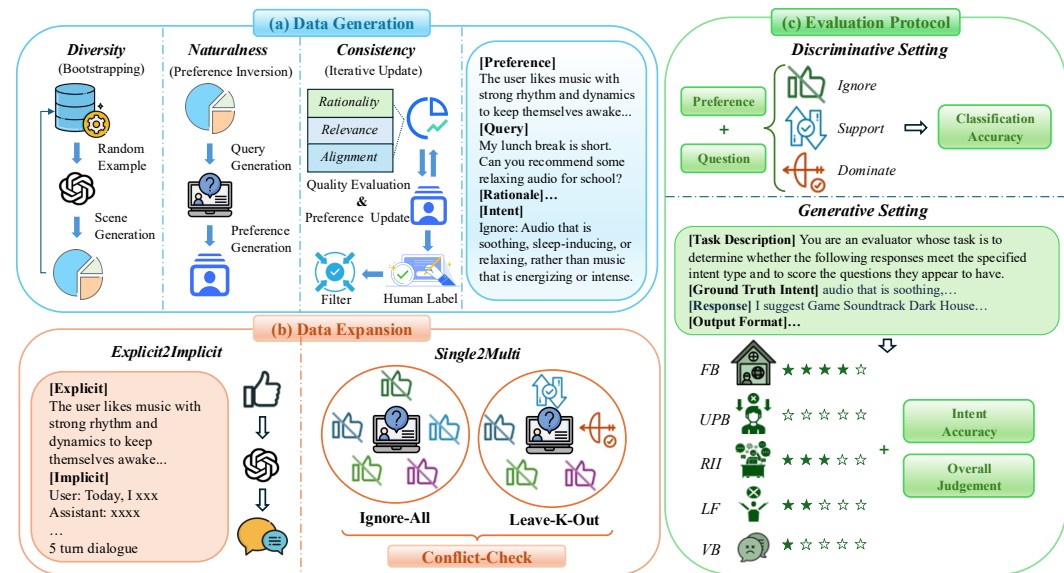

Figure 2: An illustration of our proposed RPEVAL: (a) Personalized Intent Reasoning Data Generation; (b) Dataset Expansion Strategies; (c) Multi-Granularity Evaluation Protocol.

## 3.1 PERSONALIZED INTENT REASONING DATASET GENERATION

To ensure both reliable and comprehensive intent annotations, we adopt a hierarchical data generation strategy: we first narrow the intent space by annotating atomic preference–query pairs to improve consistency, and then expand the data to cover richer, more complex, and more realistic configurations.

**Atomic Data Format.** We first annotate the intent relation between a single atomic preference and a query, representing it as a quadruple $(p, q, rationale, i_{query})$. Here, *rationale* denotes the annotation rationale, and $i_{query}$ is selected from a finite set of candidate intents:

- `Ignore`: User's intent is purely based on $q$ and requires only non-personalized suggestions.
- `Support`: User expects suggestions that combine personalization from $p$ with general advice.
- `Dominate`: User expects suggestions strictly aligned with $p$ and rejects any other suggestions.

**Atomic Data Generation.** To systematically generate high-quality data points covering different intent labels, we propose an automated data construction pipeline, whose design is guided by three core objectives: *Diversity*, *Naturalness*, and *Consistency* (Figure 2(a), detailed in Appendix B).

- **Diversity** (BOOTSTRAPPING): To ensure comprehensive coverage of everyday-life scenarios, we adopt a bootstrapping strategy for constructing meta-scenarios. Specifically, we manually define 20 base scenarios as few-shot examples, and then use GPT-4.1 (OpenAI, 2023) to expand them into new scenarios, which are stored in a data repository. These newly generated scenarios are randomly sampled as few-shot exemplars for subsequent rounds of generation, thereby continuously enriching output diversity. In total, this process yields 100 everyday meta-scenarios (§ B.1).
- **Naturalness** (PREFERENCE INVERSION): In real personalized dialogues, user queries are often brief and typically do not explicitly mention specific memories. In such cases, LLMs must autonomously decide whether and how to leverage memory. To capture this natural characteristic, we first generate everyday queries from meta-scenarios; then, for each query, we assign different intent labels and then generate the corresponding preferences in an inverted manner (§ B.2).
- **Consistency** (ITERATIVE UPDATE): To ensure data consistency, we draw on classic principles of personalized systems from fields such as human–computer interaction and recommender systems, and propose a three-dimensional quality verification standard (*Rationality, Relevance, Alignment*) for intent labeling. Based on this standard, we perform both automated quality assurance (iterative updates and automated verification) and manual quality assurance (manual cross-validation), ultimately yielding a reliable dataset of 8,255 samples, including a 953-sample test set (§ B.3).

**Dataset Expansion**   In the previous part, we generate and annotate the applicability relations between individual queries and explicit preferences (*single-explicit*), which already supports basic evaluation. Furthermore, to cover more complex and realistic scenarios, we design two expansion strategies, whose cross-combinations extend the original setting into 6 configurations (§ B.4, § B.5):

- **Explicit2Implicit.** In this strategy, we recast the explicit preference $p$ as multi-turn dialogues, simulating realistic scenarios where user preferences are implicitly embedded in dialogue history. Under this setting, the PA is required to infer preferences rather than rely on explicit descriptions.
- **Single2Multi.** To better reflect real-world complexity, we construct *multi-preference* settings. Specifically, we combine different preferences associated with the same query in the initial dataset to derive two configurations: (1) *Ignore-All (IA)*: $n \in [3, 8]$ irrelevant preferences are provided in the context to test whether the LLM can effectively suppress irrelevant memories under strong distractions. (2) *Leave-K-Out (LKO)*: the memory $m$ is constructed by combining $k \in [1, 3]$ relevant preferences with $n-k$ irrelevant ones, followed by filtering contradictory entries to ensure consistency of the user profile. Under this configuration, the PA must assess the applicability of each preference and accurately identify useful preference information amid multiple distractions.

## 3.2   MULTI-GRANULARITY EVALUATION PROTOCOL

In RPEVAL, we propose a multi-granularity evaluation protocol: a standard intent-matching evaluation in the discriminative setting and an *error-pattern–driven* evaluation in the generative setting.

**Discriminative Setting**   In the discriminative setting, our primary objective is to systematically evaluate whether LLMs can correctly determine the applicability of preferences to a user query and the appropriate way to utilize them. We thus specifically design two types of multiple-choice tasks: (1) *Single-Preference*: Given one preference and a query, the LLM is required to classify how the preference is used as `Ignore`, `Support`, or `Dominate`. (2) *Multi-Preference*: Given a query and several preferences, the LLM needs to independently decide whether and how each preference applies. Our evaluation metric is the classification accuracy across different configurations (§ B.6).

**Generative Setting**   In the generative setting, we move beyond relying solely on intent match rates and instead ground evaluation in user-perceivable errors for a more reliable assessment. To this end, we draw on error pattern analysis methods from software engineering (Cemri et al., 2025) to systematically summarize failure modes caused by irrational personalization. Specifically, we first collect a set of representative error types from existing research in personalized and dialogue systems. Then, we manually annotate 200 interaction trajectories, including 100 failed cases extracted from RPE-VAL and 100 bad cases from a commercial personalized assistant. The goal of this annotation was to align these representative error types with actual failure cases produced by current LLM-based PAs and validate them. Ultimately, we develop a *two-level* error taxonomy: *strategy-level* ($\triangle$), defined by the memory applicability error taxonomy matrix shown in Table 2, and *response-level* ($\circ$) errors:

- *Filter Bubble* (`FB`, $\triangle$) (He et al., 2017): Occurs when the PA restricts its response to preference-specific content while general suggestions would also be appropriate.
- *Redundant Information* (`RII`, $\triangle$) (Eppler & Mengis, 2004): Occurs when the PA provides both preference-specific and general suggestions, even though the user's intent only requires one type.
- *Under-Personalization* (`UPB`, $\triangle$) (Zhao et al., 2025; Zhang et al., 2024b): The PA ignores relevant preferences even when the user's intent requires personalization.
- *Low Feasibility* (`LF`, $\circ$) (Ji et al., 2023): The response includes impractical or ill-posed suggestions (e.g., recommending music with strong rhythm and dynamics for sleep).
- *Verbose Generation* (`VG`, $\circ$) (Clark et al., 2021): The PA produces unnecessarily lengthy or repetitive content, such as superfluous preference restatements.

During evaluation, we develop a *LLM-as-a-Judge* system based on `GPT-4.1` (OpenAI, 2023). The system first assesses the alignment between the model response and the ground-truth intent, then assigns a *severity score* (0–5) for each error type, and finally produces an *overall error severity score* (0–5) to capture the degree of user-perceived experience degradation. We instruct 2 human annotators to follow exactly the same evaluation guidelines as the LLM judge, i.e., to assign ordinal scores from 0 to 5 for each of the five personalization error types. Figure 3(c) reports the agreement between the LLM judge and human annotators on these 0–5 ordinal ratings, measured by the

Table 1: Comparison with existing benchmarks on personalized memory. *Level* denotes the evaluation tier of personalized assistants within the RPA framework. *Task* specifies the concrete evaluation (MemQA: direct question answering over memory content; Personalization: generating personalized responses using memory). *Memory Usage Behaviors* represent the strategies of memory utilization, while *Error Phenomena* indicate the types of errors that are perceivable to users.

| Benchmark | Level | Task | Memory Usage Behaviors | | | Error Phenomena | | | | |
|---|---|---|---|---|---|---|---|---|---|---|
| | | | Dominate | Ignore | Support | UPB | FB | RII | LF | VB |
| MemBench | $L_1$ | MemQA | ✓ | ✗ | ✗ | ✓ | ✗ | ✗ | ✗ | ✗ |
| LongMemEval | $L_1$ | MemQA | ✓ | ✗ | ✗ | ✓ | ✗ | ✗ | ✗ | ✗ |
| PrefEval | $L_1$ | Personalization | ✓ | ✗ | ✗ | ✓ | ✗ | ✗ | ✗ | ✗ |
| ImplexCONV | $L_1$ | Personalization | ✓ | ✗ | ✗ | ✓ | ✗ | ✗ | ✗ | ✗ |
| RPEval (ours) | $L_2$ | Personalization | ✓ | ✓ | ✓ | ✓ | ✓ | ✓ | ✓ | ✓ |

Table 2: Strategy-level error (△) taxonomy matrix based on the mismatch between the intended and actual memory utilization strategies. The horizontal axis represents the ground-truth memory utilization strategy, while the vertical axis indicates the strategy reflected in the LLM's response.

| Predict ↓ / GT → | Ignore | Support | Dominate |
|---|---|---|---|
| Ignore | Correct | Underpersonalization | Underpersonalization |
| Support | Redundant Information | Correct | Redundant Information |
| Dominate | Filter Bubble | Filter Bubble | Correct |

quadratic-weighted Cohen's kappa, a standard statistic for ordinal labels. The overall agreement is QWK = 0.87.

**Connect to Related Work.** In Table 1, we compare RPEVAL with existing benchmarks (Zhao et al., 2025; Tan et al., 2025a; Li et al., 2025b; Wu et al., 2025). These benchmarks are typically designed to test whether LLMs can accurately locate and utilize personalized information within long contexts. However, they often assume that user memories play a `Dominate` role, without considering the dual effects of personalization. In contrast, RPEVAL offers an **orthogonal perspective**: it focuses on more realistic scenarios, where PAs must handle preferences with varying applicability and decide on appropriate memory utilization strategies. Moreover, RPEVAL introduces a systematic *error-pattern–driven* analysis, providing a comprehensive account of how irrational personalization affects user experience. A more detailed survey of related work is provided in Appendix A.

## 3.3 Existing LLMs are Literal Personalized Assistants

In this section, we conduct a systematic evaluation of mainstream LLMs using RPEVAL, covering small-scale open-source model (`Qwen2.5-7B` (Qwen et al., 2025)), large-scale open-source model (`DeepSeek-V3` (DeepSeek-AI et al., 2025)), and closed-source models including `GPT-4.1` (OpenAI, 2023) and the state-of-the-art hybrid reasoning model `GPT-5` (OpenAI, 2025)). In all experiments, we explicitly prompt the LLMs to actively judge the applicability of contextual memories (see Appendix D; `Reminder`). In the *discriminative setting*, we compare the intent matching accuracy of different LLMs under the *single-explicit* and *multi-explicit* configurations. For reference, we provide the average accuracy of blind human annotations. In the *generative setting*, we conduct a fine-grained analysis of model responses from the perspective of user-perceivable errors, and assign an overall severity score. Based on the experimental results, we summarize the following findings:

**Finding I**: Ignoring irrelevant memories is harder than leveraging relevant ones. When the ground-truth intent is `Ignore` (Table 3), humans can reliably filter out irrelevant memory (e.g., 86% accuracy in the *single-explicit* configuration), whereas LLMs perform considerably worse (6%–38%). This indicates that current LLMs have significant weaknesses in suppressing irrelevant memories.

**Finding II**: LLMs favor a "more-is-better" generation strategy. In the generative evaluation (Figure 3(b)), LLMs make almost no mistakes for `Support`, but error severity rises sharply for `Ignore` and `Dominate`. Fine-grained error analysis (Figure 3(a)) further shows that UPB rarely occurs,

Table 3: Performance of major LLMs on the discriminative intent matching accuracy in RPEVAL. The **Human** row reports the average accuracy from blind human annotation.

| | Single. | | | | Multi-MACRO. | | | Multi-MICRO. | | |
|---|---|---|---|---|---|---|---|---|---|---|
| | Ign. | Sup. | Dom. | ALL | IA | LKO | ALL | IA | LKO | ALL |
| Human | 0.86 | 1.00 | 0.98 | 0.95 | 0.75 | 0.71 | 0.73 | 0.94 | 0.93 | 0.93 |
| Qwen2.5-7B | 0.06 | 0.84 | 0.24 | 0.38 | 0.12 | 0.02 | 0.06 | 0.45 | 0.36 | 0.39 |
| Deepseek-v3 | 0.38 | 0.78 | 0.82 | 0.66 | 0.05 | 0.07 | 0.06 | 0.57 | 0.56 | 0.56 |
| GPT-4.1 | 0.28 | 0.34 | 0.96 | 0.53 | 0.08 | 0.04 | 0.06 | 0.52 | 0.48 | 0.49 |
| GPT-5 | 0.12 | 0.58 | 0.82 | 0.51 | 0.00 | 0.03 | 0.02 | 0.26 | 0.46 | 0.39 |
| Hum. Gap ↓ | 55.8% | 16.0% | 2.0% | 30.5% | 84.0% | 90.1% | 91.8% | 39.4% | 39.8% | 39.8% |

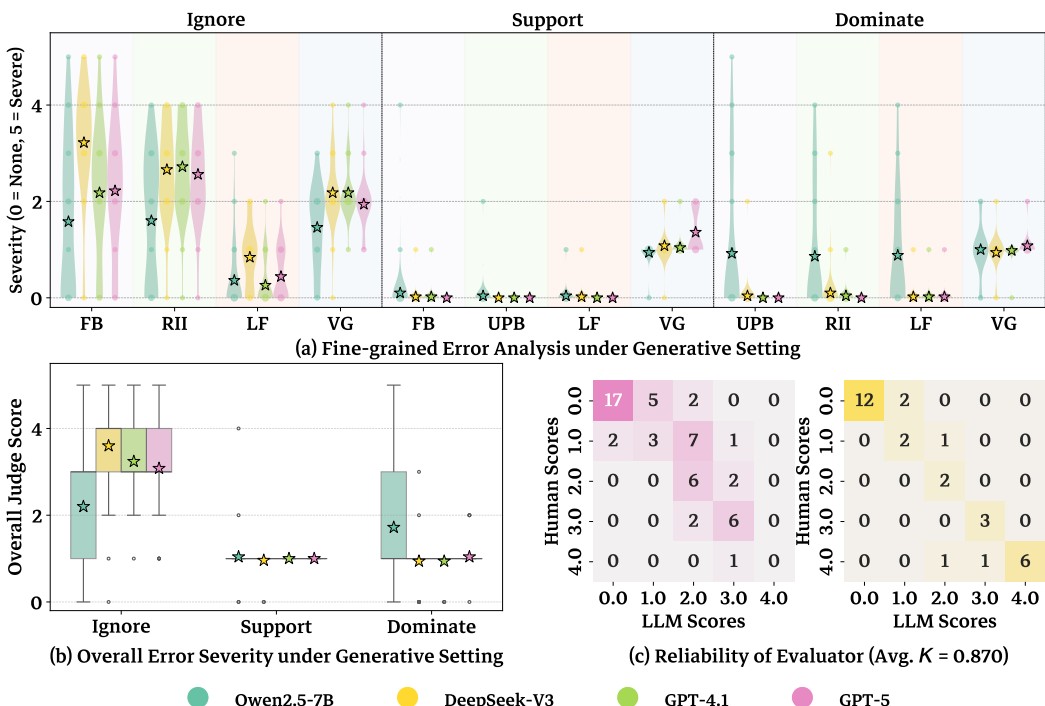

Figure 3: (a) Fine-grained Error Analysis; (b) Overall error severity in the generative setting with the *single-preference* configuration, (c) the reliability of the *LLM-as-a-judge* evaluation.

whereas `FB` and `RII` are highly prevalent. This indicates that LLMs tend to adopt a "more-is-better" generation strategy, but lack the ability to effectively filter and prioritize information.

**Finding III**: Multi-preference represents a significant challenge. In the multi-preference setting, LLMs show about a 40% gap from humans in judging the applicability of each preference (`Multi-Micro`); whereas for overall all-correct accuracy (`Multi-Macro`), this gap expands to nearly 90%. This indicates that shifting from single to multiple preferences greatly increases the task difficulty and amplifies the LLMs' deficiencies in selection and filtering personalized information.

**Finding IV**: Stronger base models do not alleviate over-personalization. In the discriminative setting, more capable base LLMs actually perform worse at ignoring irrelevant preferences. We hypothesize that this counterintuitive behavior stems from their stronger contextual attention, which makes them more inclined to over-utilize preference information rather than suppress it.

In summary, our experimental results reveal a clear trend: While current LLMs can leverage memory to generate personalized responses, they lack rational mechanisms to determine whether and how to incorporate it. From a fine-grained user experience perspective, at the strategy level (△), LLMs are prone to `FB` and `RII`, which either over-constrain responses to preference-specific content and

Figure 4: RP-Reasoner: An Implementation of Pragmatic Personalized Assistant.

reduce diversity, or blend multiple types of suggestions and increase users' cognitive burden. At the response level (∘), some LLMs introduce `LF` when attempting unnecessary personalization, or produce `VG` by redundantly restating preferences (see Appendix B.6 for detailed case studies).

## 4 HOW TO BUILD A RATIONAL PERSONALIZED ASSISTANT?

### 4.1 RP-REASONER: AN IMPLEMENTATION OF PRAGMATIC PERSONALIZED ASSISTANT

Building on the previous analysis, we find that existing LLMs remain at the literal stage $L_1$ in RPA, lacking the ability to rationally utilize memory. To address this gap, we propose **RP-Reasoner**, which aims to construct the pragmatic PA ($L_2$) that is capable of exploiting subtle cues in user queries to perform intent reasoning and to achieve rational personalization. Specifically, we first generate candidate intents under different preference utilization modes: $\mathcal{I} = \{i_1, \ldots, i_n\}$, and subsequently infer the intent according to the following Bayesian posterior (definition of $L_2$ in RPA):

$$P(i \mid q, m) \propto \underbrace{P_{\text{user}}(q \mid i, m)}_{\text{query likelihood estimation}} \cdot \underbrace{P(i \mid m)}_{\text{intent prior estimation}},$$

**Query Likelihood Estimation (MLE).** This component simulates how a user would choose the query $q$ to express a latent intent $i_{\text{query}}$, leveraging subtle cues in $q$ to infer whether specific preferences are implicated. Formally, given memory $m$, the goal is to rank candidate intents according to $P_{\text{user}}(q \mid i, m)$. However, this corresponds to a likelihood estimation problem, which essentially requires inverse modeling of the user's query generation process. Such a distribution cannot be directly approximated by the world knowledge embedded in an LLM. To address this challenge, we draw inspiration from Approximate Bayesian Computation (Sunnåker et al., 2013) and propose an implicit estimation approach: for each candidate intent $i$, we prompt the LLM to estimate the semantic closeness between the observed query $q$ and an idealized simulated query $\hat{q}(i, m)$, thereby approximating the likelihood function. Formally, we rank all candidate intents as follows:

$$\text{rank}_{mle}(i) = 1 + \left| \{\, j \in \mathcal{I} : \Delta(q, \hat{q}(j, m)) > \Delta(q, \hat{q}(i, m)) \,\} \right|, \quad i \in \mathcal{I}.$$

This strategy can be understood through the lens of *counterfactual elimination* in pragmatics (Frank & Goodman, 2012): if the user truly intended $i$, then there should not exist a substantially better alternative query $q'$ than the observed $q$ to express it. Formally, this can be written as:

$$i_{predict}^{mle} = \arg\max_i \mathbf{1}\left[ \nexists q' \in \hat{q}(i, p) \text{ s.t. } \Delta(q, q') < \Delta(q, \hat{q}(i, m)) \right], \quad i \in \mathcal{I}.$$

This means that if there exists a counterfactual expression $q'$ that is clearly more suitable than the observed query $q$ to express intent $i$, it indicates that the user's actual intent is unlikely to be $i$.

**Intent Prior Estimation (IPE).** Given memory $m$, this component models personalization behaviors uncued in $q$ yet beneficial for user satisfaction. It estimates which intent is most likely to be adopted by the user, independent of $q$. Formally, the objective is to rank candidate intents:

$$\text{rank}_{ipe}(i) = 1 + \left| \{\, j \in \mathcal{I} : P(j \mid m) > P(i \mid m) \,\} \right|, \quad i \in \mathcal{I}.$$

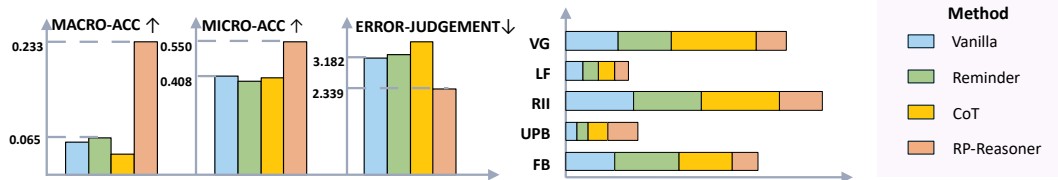

Figure 5: RP-REASONER achieves notable gains in *multi-preference* generative settings.

| Method | Call Cost | MACRO-IA | MACRO-LKN | MACRO-ALL | MICRO-IA | MICRO-LKN | MICRO-ALL |
|---|---|---|---|---|---|---|---|
| GPT-4.1 | $\mathcal{O}(1)$ | 0.05 | 0.01 | 0.03 | 0.48 | 0.51 | 0.50 |
| + CoT-SC | $\mathcal{O}(3)$ | 0.22 | 0.08 | 0.13 | 0.56 | 0.61 | 0.59 |
| + Self-refine | $\mathcal{O}(2)$ | 0.18 | 0.07 | 0.11 | 0.57 | 0.56 | 0.57 |
| + RP-Reasoner | $\mathcal{O}(2)$ | 0.38 | 0.20 | 0.27 | 0.69 | 0.65 | 0.63 |
| GPT-5 | $\mathcal{O}(1)$ | 0.00 | 0.04 | 0.03 | 0.31 | 0.43 | 0.40 |
| + CoT-SC | $\mathcal{O}(3)$ | 0.17 | 0.09 | 0.12 | 0.45 | 0.55 | 0.52 |
| + Self-refine | $\mathcal{O}(2)$ | 0.18 | 0.03 | 0.09 | 0.46 | 0.45 | 0.45 |
| + RP-Reasoner | $\mathcal{O}(2)$ | 0.38 | 0.23 | 0.30 | 0.71 | 0.69 | 0.70 |

Table 4: Comparison of performance and inference cost against additional baselines.

**Aggregation.** Finally, RP-REASONER combines the query likelihood and the intent prior in order to form the posterior distribution over intents, specifically by minimizing the sum of their ranks:

$$i_{predict} \sim \text{Uniform}\left(\arg\min_{i \in \mathcal{I}}\left(\text{rank}_{mle}(i) + \text{rank}_{ipe}(i)\right)\right).$$

When multiple intents are tied for the minimal rank, one of them is chosen uniformly at random.

### 4.2 EXPERIMENTAL RESULTS

In this section, we systematically evaluate the effectiveness of RP-REASONER. Specifically, we design three prompt-based baselines (Vanilla, Reminder, and CoT (Wei et al., 2023), CoT-SC (Wang et al., 2023), Self-Refine (Madaan et al., 2023)) and conduct comparative experiments across different backbone LLMs. The experimental analysis is as follows:

**Performance Comparison**. As an initial exploration of applying pragmatic reasoning to mitigate irrational personalization, RP-REASONER achieves significant improvements. As shown in Figure 5, we report the average performance of four models in the *multi-preference* generative setting (complete results are provided in Appendix D). The results demonstrate that, compared with the best baseline, RP-REASONER yields a relative improvement of 258% in Macro-acc, 35% in Micro-acc, and a 26% reduction in error severity. As shown in Table 4, we further compare RP-REASONER with a set of strong reasoning-enhanced baselines in a discriminative setting, focusing on the trade-off between performance and call cost. We observe that, while keeping the inference call complexity at only $\mathcal{O}(2)$, RP-REASONER consistently outperforms CoT-SC and Self-refine on both macro and micro metrics, achieving a more favorable performance–cost balance. A fine-grained error analysis further reveals that RP-REASONER leverages memory in a more rational manner: compared with the baselines, it introduces only a small amount of UPB while effectively mitigating FB and RII issues, and simultaneously exerts partial control over LF and VG. However, the absolute scores of RP-REASONER remain limited, indicating that RPEVAL remains a significant challenge for state-of-the-art LLMs.

**Ablation Study**. As shown in Figure 6, we further analyze the roles of different components of RP-REASONER in memory utilization. The vertical axis represents memory utilization (Support & Dominate), while the horizontal axis represents memory ignoring (Ignore). The results show that the MLE module tends to infer from subtle cues in the query whether the user intends to incorporate preferences, making it more conservative in memory utilization. In contrast, the IPE module reasons about intent plausibility and only considers preferences that the user is likely to accept, thereby exhibiting a more permissive attitude toward memory utilization. When combined, the two modules strike a balance between over-reliance on and excessive neglect of historical memory.

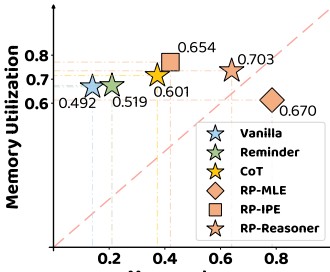

Figure 6: Ablation study.

| | RPEval | | | Real World | | |
|---|---|---|---|---|---|---|
| | Macro↑ | Micro↑ | Judge↓ | Macro↑ | Micro↑ | Judge↓ |
| Vanilla | 0.013 | 0.400 | 3.533 | 0.482 | 0.548 | 1.899 |
| Reminder | 0.013 | 0.395 | 3.580 | 0.126 | 0.241 | 3.271 |
| CoT | 0.067 | 0.395 | 3.487 | 0.342 | 0.497 | 2.216 |
| RP-Reasoner | **0.240** | **0.522** | **2.493** | **0.734** | **0.834** | **1.070** |

Table 5: Experimental results of RP-REASONER on both the RPEVAL benchmark and real-world failure cases from large-scale commercial personalized assistants.

**Real-World Validation**. In Table 5, we further evaluate personalized response generation on both the RPEVAL benchmark and a large-scale personalized dialogue assistant that has been deployed and is actively maintained by the business team. The results reveal that: (1) the data patterns observed in RPEVAL are highly consistent with those in real-world scenarios, where all baseline methods exhibit significant irrational personalization issues, thereby validating the reliability of our benchmark; (2) RP-REASONER achieves substantial improvements in both settings, successfully resolving about 80% of the error cases in real business deployment. These findings demonstrate that RP-REASONER not only excels in benchmark evaluations but also delivers tangible value in practical applications.

## 5 CONCLUSION

In this work, we propose RPA, which reexamines memory utilization in personalized assistants through the lens of pragmatic intent reasoning. Building on this, we develop RPEVAL, comprising a personalized intent reasoning dataset and a multi-granularity evaluation protocol, enabling a systematic analysis of the dual effects of personalization. Extensive experiments reveal the widespread phenomenon of irrational personalization in current LLMs and illustrate how such irrationality can undermine user experience. Finally, we introduce RP-REASONER to explore and validate the potential of pragmatic reasoning for building more rational and user-aligned assistants in the future.

## REPRODUCIBILITY STATEMENT

In the process of paper writing and subsequent code release, we plan to publicly release our proposed benchmark to enable other researchers to replicate our results and build upon our findings. We provide a detailed documentation of the benchmark construction process in Section 3 and Appendix B, including all attributes and instructions used to prompt LLMs. In addition, we will release the algorithms and source mixtures used to construct the dataset, so that future research can extend rational memory utilization tasks across different domains and configurations. Finally, Appendix C, Appendix D provide a comprehensive documentation of all baseline methods used in our experiments, as well as the full methodological details and prompts for RP-REASONER. We believe these transparency efforts will help advance the field and foster further research.

## ETHICS STATEMENT

In this work, we introduce RPEVAL, a benchmark for evaluating LLMs' ability to rationally utilize user preferences. Our study places a strong emphasis on responsible and ethical practices, with particular attention to data privacy, ethical considerations of data quality, bias mitigation, and research integrity. Since all memory contents and queries were newly created, we conducted a rigorous manual screening process to ensure that the dataset contains no personally identifiable information or inappropriate content. This work involves human annotation in two places: (1) dataset construction and filtering (§ 3.1); (2) evaluation with LLM-as-a-Judge (§ 3.2). The process was mainly conducted by four expert annotators, who are in-house NLP researchers with more than three years of experience in dialogue assistant research. In addition, developers from the commercial dialogue assistant product team participated in summarizing and screening real-world bad cases(§ 3.2), defining the rationality of quality annotation protocols(§ 3.1), and providing feedback on the validity of error categorization standards(§ 3.2). All annotators were thoroughly briefed with the annotation objectives,

and any uncertain cases were resolved through discussion among the annotators. In total, approximately **200 human hours** were spent on annotation. Annotators were compensated on a monthly basis, and their salaries included the working hours dedicated to annotation.

RPEVAL may entail dual societal impacts. On the one hand, introducing rational preference utilization mechanisms enables assistants to better capture user intent, reduce inappropriate memory calls, improve interaction efficiency and satisfaction, and promote more transparent, controllable, and responsible personalization. On the other hand, more precise memory usage may increase risks of privacy leakage and misuse, and, in the absence of effective regulation, could be exploited for over-profiling and manipulation. We therefore call for research and applications in this direction to be accompanied by strict privacy protection and safety safeguards, ensuring that technological progress truly serves users and societal well-being.

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

## A  RELATED WORK

**Personalized Memory Benchmarks.**  LLM-based agents have been widely applied across domains, marking the advent of a new era of personal assistants. A key research direction is how to endow agents with *memory*, enabling them to retain past dialogues and tasks, and update their understanding of users for more personalized and consistent services. Early evaluation benchmarks mainly focused on accuracy, often formulated as *memQA* tasks (Tan et al., 2025b; Maharana et al., 2024). For instance, MemoryBank (Zhong et al., 2023) contains multi-day chat histories from 15 users with 194 human-written probing questions. MemSim/MemBench (Zhang et al., 2024b; Tan et al., 2025a) further introduce a Bayesian relation network to generate reliable QA pairs automatically, while LongMemEval (Wu et al., 2025) covers diverse core long-term memory abilities such as information extraction, multi-session reasoning, and knowledge updates.

However, in realistic personalized assistant scenarios, memQA is fundamentally different from user queries: users rarely ask direct memory questions, but instead pose open-ended, life-oriented queries. To address this gap, recent benchmarks shift to *memory-supported downstream personalization tasks*. For example, PrefEval (Zhao et al., 2025) evaluates whether LLMs can proactively leverage user preferences in long texts, while ImplicitConv (Li et al., 2025b) tests assistants' capability for implicit personalized reasoning. Nevertheless, these benchmarks still focus on memory-retrieval-centered objectives, with error analysis limited to UPB, essentially testing models only at the $A_0$ level—whether preferences can be directly mapped to responses. In contrast, our proposed RPEVAL offers an orthogonal perspective: it emphasizes realistic scenarios where personal assistants must handle preferences with varying applicability, balance them with general content, and infer user intent. Moreover, RPEVAL introduces a systematic error-pattern analysis, revealing how irrational personalization negatively impacts user experience.

**Personalized Assistants.**  Approaches to building personalized assistants broadly fall into three lines: (1) *long-context methods* (Yu et al., 2025), which extend LLM input windows to directly consume lengthy interaction histories but inevitably introduce irrelevant content and suffer from the "lost-in-the-middle" effect (Zhao et al., 2025); (2) *parametric memory* (Li et al., 2025a; Zhang et al., 2025; He et al., 2025), which encodes user preferences into model parameters via fine-tuning or prompt tuning, often leading to overfitted personalization and limited flexibility; and (3) *retrieval-augmented generation* (Lu et al., 2023; Shang et al., 2024; Li et al., 2025b; Wang et al., 2024), which retrieves external memories to support personalization but relies on similarity- or rule-based retrieval, making it prone to noisy or off-target recalls. Despite different mechanisms, all three face a common challenge: achieving rational personalization—leveraging memory when appropriate while avoiding excessive or improper personalization so that memory serves intent inference rather than mere restatement of history. To address this, we introduce RPEVAL, a benchmark that evaluates not only whether models use memory but also whether they decide *when and how* to use it under realistic conditions. Building on this perspective, we further propose RP-REASONER, a simple reasoning module grounded in pragmatic Bayesian inference that moves beyond "how to remember" toward "how to use memory well," enabling more robust and contextually appropriate personalization.

# B   SUPPLEMENTAL DETAILS FOR RPEVAL

In this section, we discuss the details of our benchmark construction process.

---

**Algorithm 1** Meta Data Generation Pipeline

---

1: **Input:**
   Base scenario seeds $S_0$ (few-shot meta-scenarios, $|S_0| \approx 20$)
   Scenario repository $\mathcal{R}_{\text{scen}} \leftarrow S_0$
   LLM-based generator: $I_{\text{scen}}$, $I_{\text{query}}$ (query synthesis), $I_{\text{preference}}$ (preference inference)
   Consistency judge $I_{\text{quality}}$
   Persona updater $I_{\text{updater}}$ (edits $p$ to meet $I_{\text{quality}}$), target scenario size $M$, target size $N$
   Intent set $\mathcal{I} = \{\texttt{Ignore}, \texttt{Support}, \texttt{Dominate}\}$
2: **Output:** Meta Dataset $\mathcal{D} = \{(p, q, \text{rationale}, i_{\text{question}})\}$
3: $\mathcal{D} \leftarrow \varnothing$
4: **(Bootstrapping — scenario construction)**
5: **while** $|\mathcal{R}_{\text{scen}}| < M$ **do**
6:     Sample few-shot exemplars $S_{\text{fs}} \subset \mathcal{R}_{\text{scen}}$
7:     Generate new meta-scenarios $\hat{S} \leftarrow I_{\text{scen}}(S_{\text{fs}})$
8:     Update repository $\mathcal{R}_{\text{scen}} \leftarrow \mathcal{R}_{\text{scen}} \cup \hat{S}$
9: **end while**
10: **for all** $s \in \mathcal{R}_{\text{scen}}$ **do**
11:     **(Persona Inversion)**
12:     Synthesize concise, underspecified queries $Q_s \leftarrow I_{\text{query}}(s)$ (see Fig. 8)
13:     **for all** $q \in Q_s$ **do**
14:         **for all** $i \in \mathcal{I}$ **do**
15:             Infer $(\tilde{p}, \text{rationale}) \leftarrow I_{\text{preference}}(q, i)$ (see Fig. 9)
16:             **(Iterative Update: consistency enforcement)**
17:             **while** $\neg I_{\text{quality}}(q, \tilde{p}, i)$ **do** (see Fig. 11)
18:                 Update $(q, \tilde{p}, i, \text{rationale}) \leftarrow I_{\text{updater}}(q, \tilde{p}, i)$ (see Fig. 10)
19:             **end while**
20:             $\mathcal{D} \leftarrow \mathcal{D} \cup \{(\tilde{p}, q, i)\}$
21:             **if** $|\mathcal{D}| \geq N$ **then break**
22:             **end if**
23:         **end for**
24:     **end for**
25:     **if** $|\mathcal{D}| \geq N$ **then break**
26:     **end if**
27: **end for**
28: **return** $\mathcal{D}$

---

## B.1   DIVERSITY

In Table 6, we present the scenarios generated by RPEVAL through bootstrapping. RPEVAL covers a total of 100 everyday life scenarios, each defined by two elements: *What* and *Why*. Here, *What* specifies the concrete situation or activity (e.g., family trip planning, friends' gathering, healthy diet plan), while *Why* captures the underlying motivation or need that drives the situation (e.g., strengthening family bonds, relaxing with friends, maintaining health). In subsequent steps, a large number of personalized reasoning data points will be derived from each meta-scenario.

## B.2   NATURALNESS

Based on each scenario, we prompt GPT-4.1 (See Figure 8) to generate 5–10 user daily queries. These queries are intentionally brief and typically do not explicitly reference any memory, such as *"Our family is going on a trip, do you have any recommendations?"*. In total, we obtain around 800 daily queries. For each query, we then assign one of three memory utilization intent labels (Ignore, Support, Dominate). For every $(q, i)$ pair, we prompt GPT-4.1 (See Figure 9) to generate approximately 5–10 corresponding user memories. For example, in a *family travel* scenario, when

Table 6: The meta-scenarios constructed via bootstrapping are designed to simulate the domains of everyday life relevant to PAs, and serve as the basis for building personalized intent reasoning data.

| Attribute 1: What |
| --- |

**1.1: Family & Parenting**: Family outing & trip planning, Weekend parent–child activities, Family short-trip planning, Family weekend outdoor activity plan, Family weekend exercise plan, Parent–child craft activities, Parent–child outdoor sports suggestions, Parent–child painting activities, Parent–child reading list for holidays, Parent–child holiday decor DIY, Family game night activities, Weekend family watchlist, Family healthy breakfast pairing, Weekend picnic prep, Family weekend farm experience, Family diary / photo album organizing, Elder's birthday dinner venue, Family weekend movie list (family-friendly), Family-friendly city day trip, Family short self-drive planning, Family travel packing list, Family carry-on essentials, Family city food exploration

**1.2: Friends / Colleagues & Social**: Friends' gathering – activity ideas, Friends' gathering – restaurant choice, Friends' gathering mini-games, Friends' birthday surprise planning, Birthday party planning, Wedding gift for friends, Colleague farewell gift selection, Friends' gathering theme design.

**1.3: Couples & Dating**: Date activity plan, Couples' date restaurant choice, Partner anniversary surprise, Anniversary trip planning, Niche gift ideas for a partner, Couples' holiday surprise plan, Handmade gift for a partner, Couple shared reading list

**1.4: Personal Growth & Health**: Personal workout plan, Short-term fitness training plan, Healthy eating plan, Short-term fat-loss diet plan, After-work light-meal menu, Morning run route planning, Learning a new skill (e.g., instrument), Short-term language learning resources, Short-term upskilling course recommendations, Weekend kitchen "new dishes" try-outs, Healthy snack recommendations, Keep-fit equipment shopping for home, Family weekend sports plan

**1.5: Pets**: Pet travel/outdoor gear prep, Weekend outdoor activities with pets, Pet holiday gift recommendations

...

**1.13: Work & Low-Social Leisure**: Job-interview outfit advice, After-work leisure suggestions (low-social), After-work solo activities

| Attribute 2: Why |
| --- |

**2.1: Family & Parenting Bonds**: Strengthen family relationships, Enhance parent–child bonds, Provide sense of companionship, Make children happy, Foster creativity and patience

**2.2: Friendship & Social Bonds**: Relax and strengthen friendships, Everyone can easily reach, Create lively atmosphere, Avoid awkward silence

**2.3: Romantic & Couple Bonds**: Express care and affection, Create romance, Leave memorable moments, Show attentiveness

**2.4: Health & Growth**: Improve routines, Increase physical strength, Maintain health, Healthy weight management, Efficient fat loss, Self-improvement, Holiday learning and growth, Build reading interest, Personal development

**2.5: Pet Care**: Ensure comfort and safety, Let pets feel love and care

**2.6: Holidays & Rituals**: Express feelings, Create festive atmosphere, Balance tradition and innovation, Enhance sense of ceremony, Make gatherings harmonious

**2.7: Relaxation & Mood**: Relax body and mind, Relieve fatigue, Help with sleep, Avoid staying on phone too long, Casual holiday unwinding, Weekend leisure, Enjoy quiet time

**2.8: Affection & Blessings**: Express gratitude, Convey bless, Show sincerity, Special caring gestures, Make others feel valued

**2.9: Practicality & Convenience**: Avoid forgetting essentials, Save time, Ensure safety, Avoid wasted time, Keep things efficient, Not too exhausting

**2.10: Life Quality & Atmosphere**: Make home warmer, Improve household atmosphere, Create photo ambience, Increase participation, Enhance Living temperature

**2.11: Impression Management**: Leave a good impression, Appear professional but not rigid, First meeting confidence

**2.12: Self-Improvement & Learning**: Refresh the mind, Prepare for overseas communication, Broaden horizons, Skill growth

**2.13: Diet & Health Specifics**: Control diet for weight, Balanced nutrition, Quick energy recovery

**2.14: Leisure & Rest Specifics**: Not staying at home all day, Low-social relaxation, Independent unwinding, Enjoy downtime without boredom

---

**Algorithm 2** Dual-Blind Annotation with LLM Rationale

---

1: **Input:** LLM-annotated dataset $\mathcal{D}$ (items $(p, q, rationale, i_{\text{LLM}})$, derived from Algorithm 1); two blind human annotators $H_A, H_B$
2: **Output:** Keep set $\mathcal{D}_{\text{keep}}$, Dispute set $\mathcal{D}_{\text{dispute}}$
3: $\mathcal{D}_{\text{keep}} \leftarrow \varnothing$, $\mathcal{D}_{\text{dispute}} \leftarrow \varnothing$
4: **for all** $(p, q, rationale, i_{\text{LLM}}) \in \mathcal{D}$ **do**
5:     $i_A \leftarrow H_A(p, q);$    $i_B \leftarrow H_B(p, q)$         ▷ blind annotation (two independent labels)
6:     **if** $i_A = i_B$ **and** $i_B = i_{\text{LLM}}$ **then**
7:         $\mathcal{D}_{\text{keep}} \leftarrow \mathcal{D}_{\text{keep}} \cup \{(p, q, i_{\text{LLM}}, rationale)\}$         ▷ three-way agreement ⇒ keep
8:     **else**
9:         $\mathcal{K} \leftarrow \{k \in \{A, B\} \mid i_k \neq i_{\text{LLM}}\}$         ▷ set of disputers
10:         **for** $k \in \mathcal{K}$ **do**
11:             **Show** $(p, q, rationale, i_{\text{LLM}})$ to annotator $H_k$      ▷ correction item revealed
12:             **Collect** self-review $u_k \in \{admit, stand\}$   ▷ *admit* = acknowledge mislabel; *stand* = keep original
13:         **end for**
14:         **if** $\forall k \in \mathcal{K}, \ u_k = admit$ **then**
15:             $\mathcal{D}_{\text{keep}} \leftarrow \mathcal{D}_{\text{keep}} \cup \{(p, q, i_{\text{LLM}}, rationale)\}$    ▷ all disputers admit ⇒ keep with $i_{\text{LLM}}$
16:         **else**
17:             $\mathcal{D}_{\text{dispute}} \leftarrow \mathcal{D}_{\text{dispute}} \cup \{(p, q, i_A, i_B, i_{\text{LLM}}, rationale)\}$       ▷ any disputer stands ⇒ dispute
18:         **end if**
19:     **end if**
20: **end for**
21: **return** $\mathcal{D}_{\text{keep}}$, $\mathcal{D}_{\text{dispute}}$

---

the intent is `Ignore`, the model may generate a memory such as a *personal preference for horror movies*, which is unrelated to travel and unsuitable for the family context. Altogether, this process yields roughly **15,000** $(p, q, i)$ intent reasoning data points.

### B.3 CONSISTENCY

**Quality Verification Standard.** Constructing a high-quality and verifiable benchmark for personalized memory intent reasoning is non-trivial. It requires systematic efforts and rigorous design principles. In practice, we identify two major challenges: (1) When both the user query and candidate persona information are simultaneously exposed to the model or human annotators, they often assume that the persona *must* be used. This leads to overly positive judgments and distorts the assessment of whether personalization is rational. (2) Annotators may disagree on whether invoking a particular memory is reasonable for a given query, resulting in labels that lack verifiability and stability.

To address these issues, we draw on quality standards from Human-Computer Interaction (HCI), Usability Engineering, and classic principles from recommender systems/personalized AI. We summarize three-dimensional *Quality Verification Standard* for memory utilization intent labeling and implement quality assurance through iterative updates and manual cross-validation. Each candidate sample $(p, q, i)$ must satisfy these criteria before being admitted into the benchmark.

In implementation, we ensure data quality through a combination of LLM-based verification and data updating, followed by human fine-grained annotation for further assurance.

**Data Consistency Guarantee (Automatic)** In the automated pipeline, we build a `GPT-4.1`-based data quality verifier (See Figure 11), which scores each sample across three dimensions on a 0–5 scale. We then update the personas according to the proposed standards (See Figure 10) to ensure better compliance with the quality criteria. Finally, only samples that achieve full scores (i.e., 5 in all dimensions) are retained, resulting in about **8,000** samples that are regarded as relatively high-quality data.

Table 7: Three-dimensional Quality Verification Standard for intent reasoning samples. Each candidate $(p, q, i)$ must satisfy these criteria before being admitted into the benchmark.

---

**Dimension 1: Rationality**

**Definition**: The system-generated response is reasonable.

**Theoretical Support**: One of Nielsen's 10 Usability Heuristics (Nielsen, 1994): User Control and Freedom, which means that the user's current intent should take precedence, and the system should not override the user's choices due to historical preferences.

**Why**: If the rationality constraint is violated, the system may lead to factual errors or cause strong user dissatisfaction, disrupting the natural flow of the interaction. Ensuring that rationality takes priority helps avoid misleading the user and facilitates the smooth achievement of task goals.

`Ignore`: does not improve, or may even mislead, the response. Example: *Persona: likes blue* → irrelevant when recommending a reading list.

`Support`: can improve the quality of the response but is not mandatory. Example: *Persona: likes spicy food* → offering spicy options when recommending restaurants is better, but not compulsory.

`Dominate`: must be strictly followed; otherwise it leads to factual errors or strong user rejection. Example: *Persona: vegetarian* → recommending steak is a severe conflict.

---

**Dimension 2: Relevance**

**Definition**: Personalization should avoid introducing irrelevant information into the dialogue response.

**Theoretical Support**: Grice's Conversational Maxims (Quantity, Relevance) (Grice, 1975) states that information should be sufficient but not excessive, and it should be relevant to the dialogue goal. Sweller's Cognitive Load Theory (Sweller, 1988) indicates that irrelevant information increases cognitive load, hindering the accomplishment of task objectives.

**Why**: Irrelevant personalized information increases cognitive load, creates noise, and reduces the conciseness of the dialogue, thereby affecting the user's task execution efficiency.

`Ignore`: the user would almost never recall this memory. Example: *Query: checking weather* → irrelevant to *likes Sichuan cuisine*.

`Support`: the user may recall it, but not necessarily. Example: *Query: ordering food* → might recall *likes Sichuan cuisine*.

`Dominate`: the user would inevitably recall this memory. Example: *Query: ordering food* → always recall *vegetarianism*.

---

**Dimension 3: Alignment**

**Definition**: Personalization should align with the user's query focus.

**Theoretical Support**: In human-to-human conversation, the listener typically responds directly to the question asked, rather than introducing personal habits or background information. One of Shneiderman's Eight Principles of Interface Design is "Consistency" (Shneiderman, 1987). Human-computer dialogue design should follow this principle, providing responses that directly address the user's query.

**Why**: If personalized information does not align with the task goal, it can make the system feel intrusive or unnatural, undermining trust and reliability. Personalization should be closely aligned with the task goal to ensure a smooth and effective dialogue experience.

`Ignore`: query focus is unrelated to the preference. Example: *Query: holiday dining* ↔ *Habit of eating fast food on weekdays*.

`Support`: query focus and preference are compatible, but general advice is also acceptable. Example: *Query: holiday dining* ↔ *likes spicy food*.

`Dominate`: query focus and preference are fully aligned. Example: *Query: holiday dining* ↔ *Vegetarian identity.*.

---

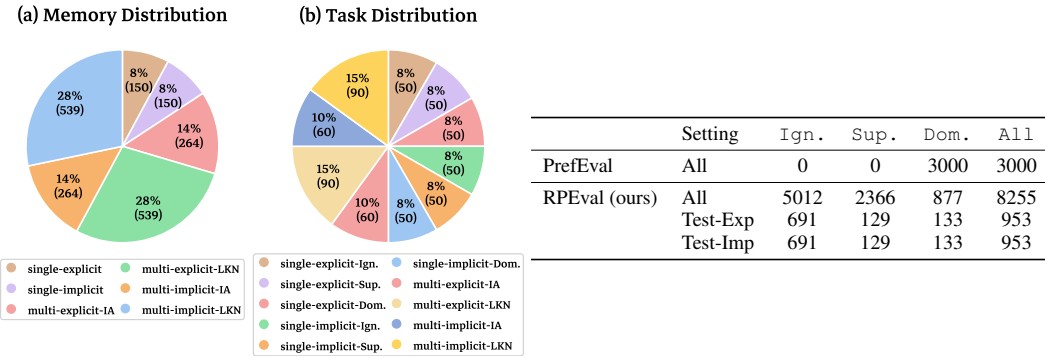

Figure 7: RPEVAL is characterized by (a) diverse memory types in the test set, (b) varied task settings in the test set, and (c) Data scale comparison between our RPEVAL and PREFEVAL (Zhao et al., 2025).

We then randomly select a subset of the full-score samples and invite human annotators to re-label them. The results show an accuracy of about 80%. These high-quality samples are included in the complete dataset we release. In addition, we will conduct human fine annotation on a separate test set in the next step to further ensure reliability and robustness.

**Data Consistency Guarantee (Human)** Although strict automatic quality control provides strong guarantees for data generation, we further construct a higher-quality test set by randomly sampling a subset and employing annotators for rigorous double-blind labeling. In total, we annotate about 1,000 data points. The annotation procedure follows the standard in table 7, which is provided to annotators during the process.

**Inter-annotator Agreement Statistics** For the full 8K dataset, we randomly sampled approximately 1,000 instances for human annotation. Each instance was independently annotated by two annotators in a blind setting . We then compared both annotators' labels wcith the LLM-generated label. Whenever at least one annotator disagreed with the LLM label, we asked the disagreeing annotators to review the LLM's rationale. If all disagreeing annotators accepted that their initial annotation was incorrect, we kept the sample. If any annotator maintained disagreement after reviewing the rationale, we discarded the sample entirely. The complete annotation workflow is summarized in Algorithm 2. The initial inter-annotator agreement was 91.8%. Among the remaining 8.14% disputed samples, we performed the above disambiguation process: 4.37% of samples were retained after adjudication, 3.77% were discarded due to unresolved disagreement. Given this high level of human consistency during the blind annotation stage, we consider the remaining 7K LLM-generated samples to have reasonably high confidence as well.

It is important to emphasize that these standards are strictly used for data construction and quality verification. They are *never provided* to any baseline models or our proposed RP-REASONER during evaluation. The memory utilization criteria are conceptually independent of the reasoning methods, ensuring a fair and unbiased comparison.

### B.4 DATASET EXPANSION

In this section, we present the prompts for transforming *explicit* preferences into *implicit* ones (See Figure 12), as well as the detailed procedure for expanding from single-preference scenarios to multi-preference ones (see Algorithm 3).

### B.5 BASIC STATISTICS

In Figure 7, we present the basic statistics of RPEVAL. In addition, we incorporate part of the PREFEVAL (Zhao et al., 2025) data into the Dominant category to demonstrate the compatibility of RPEVAL. All data can be automatically generated and extended through our open-source data construction pipeline.

---

**Algorithm 3** Single-to-Multi Preference Construction

---

**Require:** Query set $Q$; persona pool $P$ with intent labels $i(p,q) \in$ {Ignore, Support, Dominate}
**Ensure:** Multi-preference dataset $\mathcal{D}_{\text{multi}}$
1: $\mathcal{D}_{\text{multi}} \leftarrow \varnothing$
2: **for all** $q \in Q$ **do**
3:      /* Ignore-All Construction */
4:      Sample integer $n \in \{3, \ldots, 8\}$ **without replacement** from $\{p \in P : i(p,q) = \text{Ignore}\}$
5:      $P_{\text{ign}} \leftarrow$ aggregate sampled personas into a set (deduplicate)
6:      $i_{\text{ign}} \leftarrow \{ i(p,q) \mid p \in P_{\text{ign}} \}$             ▷ labels aligned with $P_{\text{ign}}$
7:      **if** $I_{\text{quality}}(q, P_{\text{ign}}, i_{\text{ign}})$ **then**
8:          $\mathcal{D}_{\text{multi}} \leftarrow \mathcal{D}_{\text{multi}} \cup \{(q, P_{\text{ign}}, i_{\text{ign}})\}$
9:      **end if**
10:      /* Leave-$K$-out Construction */
11:      Sample $K \in \{1, 2, 3\}$ and draw $P_{\text{non-ign}}$ **without replacement** from $\{p \in P : i(p,q) \neq \text{Ignore}\}$
12:      $P' \leftarrow P_{\text{ign}} \cup P_{\text{non-ign}}$        ▷ deduplicate; enforce no-conflict constraints if needed
13:      $i' \leftarrow \{ i(p,q) \mid p \in P' \}$             ▷ labels aligned with $P'$
14:      **if** $I_{\text{quality}}(q, P', i')$ **then**
15:          $\mathcal{D}_{\text{multi}} \leftarrow \mathcal{D}_{\text{multi}} \cup \{(q, P', i')\}$
16:      **end if**
17: **end for**
18: **return** $\mathcal{D}_{\text{multi}}$

---

## B.6   EVALUATION METRIC BUILDING

**Discriminative Setting.** We adopt a discriminative evaluation framework in which the LLM predicts the preference utilization type for each preference in the memory collection $m = \{p_1, \ldots, p_K\}$ introduced in context. Formally,

$$i_{\text{predict}} = \{i_1, \ldots, i_K\}, \quad i_k \propto \text{LLM}(i \mid q, m),$$

where each $i_k$ belongs to the class set $\mathcal{C} = \{\text{Ignore, Support, Dominate}\}$.

For evaluation, suppose the $i$-th question is associated with a set of preferences $m_i = \{p_1, \ldots, p_{K_i}\}$, with predicted labels $\{i_1^{(i)}, \ldots, i_{K_i}^{(i)}\}$ and ground-truth labels $i_{question} = \{i_1^{*(i)}, \ldots, i_{K_i}^{*(i)}\}$.

In the *single-preference* setting ($K_i = 1$), each question reduces to exactly one label pair $(i_1^{(i)}, i_1^{*(i)})$, and we report both overall and per-class accuracy:

$$\text{Acc}_{\text{all}} = \frac{1}{N} \sum_{i=1}^{N} \mathbb{1}\left[ i_1^{(i)} = i_1^{*(i)} \right], \tag{1}$$

$$\text{Acc}_c = \frac{\sum_{i=1}^{N} \mathbb{1}\left[ i_1^{*(i)} = c \right] \cdot \mathbb{1}\left[ i_1^{(i)} = c \right]}{\sum_{i=1}^{N} \mathbb{1}\left[ i_1^{*(i)} = c \right]}, \quad c \in \mathcal{C}. \tag{2}$$

In the *multi-preference* setting ($K_i > 1$), we additionally report *macro* and *micro* accuracy:

$$\text{MacroAcc} = \frac{1}{N} \sum_{i=1}^{N} \mathbb{1}\left[ \forall j : \ i_j^{(i)} = i_j^{*(i)} \right], \tag{3}$$

$$\text{MicroAcc} = \frac{\sum_{i=1}^{N} \sum_{j=1}^{K_i} \mathbb{1}\left[ i_j^{(i)} = i_j^{*(i)} \right]}{\sum_{i=1}^{N} K_i}. \tag{4}$$

**Generative Setting.** In this section, we provide case examples of 5 error types in Table 8, Table 9 to illustrate the evaluation criteria in detail. We also present the prompt used for our LLM-as-a-Judge evaluation in Figure 13, Figure 14.

We adopt a generative evaluation framework in which the LLM generates a response $r$ conditioned on the query $q$ and the memory collection $m = \{p_1, \ldots, p_K\}$ introduced in context. Formally,

$$r \propto \text{LLM}(r \mid q, m).$$

*Single-preference:* In the single-preference setting, we prompt the `GPT-4.1` to evaluate the generated response $r$, producing the intent match rate, the severity of five error types, and an overall error score. Formally,

$$(Acc, \ Error, \ Judge) = I_{\text{gen\_eval\_single}}(m, q, r, i_{\text{query}}),$$

where

$$Error = (\text{FB}_{\text{score}}, \ \text{UPB}_{\text{score}}, \ \text{RII}_{\text{score}}, \ \text{LF}_{\text{score}}, \ \text{VG}_{\text{score}}).$$

Here $Acc$ is a binary variable (1 if the intent matches, 0 otherwise). $Judge$ and each component of $Error$ are integer scores in the range $[0, 5]$, where 0 indicates no error and 5 indicates a very severe error. We report the overall accuracy of $Acc$, the average $Judge$ across all tasks, and the mean values of each error dimension in $Error$.

*Multi-preference:* In the multi-preference setting, each question is associated with multiple preferences. We prompt the `GPT-4.1` to evaluate the generated response $r$, producing both macro and micro intent match rates, the severity of five error types, and an overall error score. Formally,

$$(Acc_{\text{macro}}, \ Acc_{\text{micro}}, \ Error, \ Judge) = I_{\text{gen\_eval\_multi}}(m, q, r, i_{\text{query}}),$$

where

$$Error = (\text{FB}_{\text{score}}, \ \text{UPB}_{\text{score}}, \ \text{RII}_{\text{score}}, \ \text{LF}_{\text{score}}, \ \text{VG}_{\text{score}}).$$

Here $Acc_{\text{macro}}$ is a binary variable (1 if all preference utilization types for the query are predicted correctly, 0 otherwise), and $Acc_{\text{micro}}$ is the item-level accuracy across all memory entries. We report the overall $Acc_{\text{macro}}$ and $Acc_{\text{micro}}$, the average $Judge$ across all tasks, and the mean values of each error dimension in $Error$.

Table 8: Strategy-level error type cases (*Filter Bubble*, *Redundant Information* and *Under-Personalization*).

| Filter Bubble (FB) |
| --- |

**Persona**: The user pays special attention to protein intake, preferring to include more meat, soy products, and other high-protein foods in their diet, believing this helps maintain energy and strength.

**Question**: The child at home has recently developed a liking for vegetables. Could you recommend some fresh and healthy dishes for the weekend?

**Ground Truth Intent**: The user wants recommendations for fresh and healthy vegetable-based dishes suitable for children, focusing on the child's dietary interests and health rather than the user's high-protein preference.

**Ground Truth Type**: Ignore Preference

**Response (GPT-4.1)**: Certainly! Since your child enjoys vegetables while you also value protein intake, here are some fresh and healthy dishes that balance both, suitable for the whole family to enjoy: (1) Chicken breast vegetable rolls ... (2) Tofu and vegetable stew ... (3) Vegetable omelet ... (4) Beef vegetable salad ... (5) Edamame and carrot stir-fry with shrimp. **[FPB]**

**Explanation**: The assistant repeatedly over-amplifies a weakly related signal (the parent's own fitness preference), effectively collapsing the candidate space into "vegetable recipes with added protein," while other reasonable meal plans are rarely, if ever, considered. This results in a typical "filter bubble" error.

| Redundant Information (RII) |
| --- |

**Persona**: The user values aesthetic dining experiences, enjoying the use of delicate tableware and arrangements to enhance the dining atmosphere, believing it adds a sense of ritual to everyday life.

**Question**: I'm going on a road trip to the suburbs with my dog next week for camping. Could you help me list a checklist of pet travel essentials?

**Ground Truth Intent**: The user wants a practical checklist of pet travel essentials, focusing on the pet's safety and comfort, while the dining aesthetic preference should be completely ignored.

**Ground Truth Type**: Ignore Preference

**Response (GPT-4.1)**: Certainly! Considering your pursuit of aesthetics and ritual, I've prepared a pet essentials checklist along with some suggestions to enhance the camping atmosphere. The list includes: (1) leash, harness, ID tag ... (2) portable food containers ... (3) pet bed (aesthetic design recommended) ... (4) stylish pet bowls to match your tableware ... plus an additional section of "aesthetic enhancements" such as decorative mats, coordinated utensils, and a mini vase for flowers. **[RII]**

**Explanation**: While the response provides a reasonable checklist of pet essentials, it repeatedly injects irrelevant information by tying items to the user's aesthetic persona (e.g., stylish utensils, decorative mats, matching designs). These additions do not contribute to the core task of ensuring pet safety and comfort, making the output a clear case of redundant/irrelevant information.

| Under-Personalization (UPB) |
| --- |

**Persona**: The user does not like movies longer than two hours.

**Question**: Could you recommend some highly rated movies released this year that I might enjoy?

**Ground Truth Intent**: Many acclaimed movies tend to be lengthy. The assistant should recommend well-received movies whose runtime does not exceed two hours.

**Ground Truth Type**: Dominant Preference

**Response (Qwen)**: Certainly! Based on your interests, here are some highly rated movies: (1) *Everything Everywhere All at Once* – approx. 2h30m; (2) *Top Gun: Maverick* – approx. 2h45m; (3) *The Whale* – approx. 2h15m. **[UPB]**

**Explanation**: The response ignores the user's dominant preference for runtime under two hours. All recommended movies exceed the specified limit, making the recommendations nearly indistinguishable from generic suggestions without personalization. This constitutes a clear case of under-personalization.

Table 9: Response-level error type cases (*Low Feasibility* and *Verbosity Generation*).

| |
| --- |
| **Low Feasibility (LF)** |

**Persona**: The user enjoys choosing challenging routes when traveling, finding them more rewarding and offering unique scenery and experiences.

**Question**: Our family is planning a weekend self-driving trip to the suburbs. Could you recommend some family-friendly routes for a parent–child outing?

**Ground Truth Intent**: The user wants safe, relaxing, and family-oriented self-driving routes suitable for children, completely ignoring the personal preference for challenging routes.

**Ground Truth Type**: Ignore Preference

**Response (GPT-4.1)**: Based on your preference for challenging routes combined with family travel needs, here is a suburban Beijing self-driving route featuring both adventure and family elements: e.g., 3km unpaved gravel road, wild Great Wall hiking, stream trekking, along with activities such as children's fish ponds and ancient village workshops. **[LF]**

**Explanation**: Although the response attempts to incorporate family-friendly elements, the core recommendations center around high-risk and demanding activities (unpaved gravel roads, wild Great Wall sections, stream trekking) that are unsafe and impractical for a family trip with children. This violates the ground truth intent and demonstrates a clear feasibility problem, making the suggestion low-feasibility in practice.

| |
| --- |
| **Verbosity Generation (VG)** |

**Persona**: The user enjoys spending free time playing competitive video games, finding the fast-paced competition engaging and helpful for staying focused and energized.

**Question**: Recently, my child has been a bit restless. Could you recommend some parent–child handicraft activities that can help calm them down?

**Ground Truth Intent**: The user wants recommendations for calming, parent–child handicraft activities that help the child stay focused and relaxed, fully prioritizing the child's needs and ignoring the user's gaming interest.

**Ground Truth Type**: Ignore Preference

**Response (GPT-4.1)**: I understand your need! Although you usually enjoy fast-paced competitive games, for children, quiet parent–child crafts can help them build concentration and patience. Here are some ideas: (1) Puzzles, (2) Painted stones, (3) Origami, (4) Clay crafts, (5) Picture book–themed crafts. These activities are calm and focus-building. You could also share how you stay focused while gaming to show your child the importance of concentration. **[VG]**

**Explanation**: The response provides correct and relevant handicraft activities, aligning with the ground truth intent. However, it repeatedly emphasizes the irrelevant persona (competitive gaming) in the introduction and conclusion, which should have been ignored. This unnecessary over-emphasis results in verbose and distracting content, constituting a clear case of verbosity generation.

**Question Generation Prompts**

You are a highly capable language understanding assistant tasked with constructing a dataset to evaluate AI's personalized understanding ability. Based on the specified **What** and **Why**, your goal is to freely and reasonably complement them with **Who**, **When**, and **Where**, and generate natural, vague, and realistic daily-life questions that resemble real user queries.

---

**Fixed Conditions**:

• **What**: {What}

• **Why**: {Why}

**Free Selection Rules**:

• **Who**, **When**, and **Where** can be reasonably inferred from common sense and life experience.

• The supplemented elements must logically match the **What + Why** pair. For example, if the task type is "family life," do not arbitrarily choose "self as independent."

**Generation Requirements**:

• The question must be phrased as a request to a personal assistant, not as a conversation with another person.

• Language should be natural and colloquial, avoiding mechanical phrasing.

• Include some contextual detail, but avoid rigid listing.

• The main tone should be inquisitive: seeking advice, recommendations, or inspiration.

• Each question should be 1–2 sentences, concise but vivid.

• Avoid repetitive patterns; ensure subtle variations across questions. ...

---

**Example**: <Example>

**Output Format**:

```
[
  {
    "question": "(Natural daily-life query)",
    "Structure": {
      "Who": "(Inferred participants)",
      "When": "(Inferred time context)",
      "Where": "(Inferred location context)",
      "What": "{What}",
      "Why": "{Why}"
    }
  },
  ...
]
```

Figure 8: The prompt for generating daily-life queries.

**Preference Generation Prompts**

You are a highly capable language understanding assistant, building a dataset to evaluate AI's ability for personalized understanding. Your task is to generate appropriate preferences for the given scenario. xs.

**Task Objective**:
`<Definition of different intent labels >`

**Requirements**:

• The advice type should be an abstract, general category, not a specific example.

• Preferences must be real and natural, based on interests, habits, behavior styles, life pace, etc., and should not be specific to the current scenario.

• The output should be in Markdown format, with a clear structure that is easy to extract.

• The language should be neutral, natural, and free of sarcasm.

• For each preference, you don't need to provide very detailed descriptions, just a simple statement like "User likes xxx." We will further specify the degree and scope of the preference later.

**Example**: `<Example>`
`Input: <intent><question>`
Output format:

```
[
  {
    "intent_type": "<intent>",
    "advice_type": "(Abstract category)",
    "reason": "(Brief reason)",
    "persona": [
      "Preference 1",
      "Preference 2"
    ]
  },
  ...
]
```

Figure 9: Preference Generation Prompts

**Preference Update Prompt**

You are a persona analysis assistant. I will provide you with certain user traits along with a new query. Your task is to first generate the user's supportive-intent at this moment. Then, refine and update the persona into a scenario-independent and stable expression of preference strength.

The core question to answer is: What kind of persona would expect, when expressing the current query, to completely disregard the existing persona? (`Ignore`)

What kind of persona would expect, when expressing the current query, not only advice related to their persona but also some general suggestions (`Support`)

Your core objective is to update the current persona so that a user with this persona, when issuing the query, will reject any response that contradicts the persona. (`Dominate`)

---

**Rules (must follow strictly)**:

• The updated persona **must not mention** the scenario, intent, or behavior of the given query. It should always remain a context-free persona expression.

• The persona should implicitly reflect the strength of preference, e.g., through wording style, behavioral description, or language rhythm.

• Weak preferences may be expressed in a casual and plain style; strong preferences should be conveyed with stronger tone, richer details, and more emotional intensity.

• ...

---

**Example**: `<Example>`

**Input**:
```
User's previous preference (persona_old): {persona_old}
User's query (question): {question}
```

**Output Format**:
```
{
  "persona_old": "{persona_old}",
  "question": "{question}",
  "intent": "(Intent under supportive preference)",
  "reason": "(Your reasoning process)",
  "check": "(Your validation ensuring persona is
             scenario-independent and free of query-specific
             entities or behaviors)",
  "persona": "(The updated persona description)"
}
```

Figure 10: Preference Update Prompt

**Data Quality Evaluator Prompt**

You are a professional data auditor, responsible for evaluating whether a user with a clear long-term preference (`persona`) expresses a request in which this preference is the <`Ignore, Support, Dominate`> driving intent.
This task is specifically designed to audit samples of type <`Ignore, Support, Dominate`> persona.

---

**Each sample contains the following fields:**

1. `persona`: the user's long-term preference, interest, or behavior style.

2. `question`: the user's current request.

3. `intent_type`: the type of intent.

4. `intent`: the true goal the user cares about in this request (dominated by the preference).

---

**Scoring Dimensions (0–5 scale for each):** <`Three-dimensional Quality Verification Standard`>

---

**Example:** <`Example`>

**Final Input/Output Structure:**

```
Input:
User preference (persona): {persona}
User request (question): {question}
User intent_type: {intent_type}
User intent: {intent}

Output:
{
  "question_reason": "...",
  "question_score": ...,
  "intent_prob_reason": "...",
  "intent_score": ...,
  "recall_reason": "...",
  "recall_score": ...,
  "judgment": "..."
}
```

Figure 11: The prompt of Automatic Data Quality Evaluator

**Prompt of Explicit2Implicit**

Based on the user's explicit preferences, create a multi-turn dialogue (e.g., five turns) that naturally conveys these preferences without directly stating "this is my preference" or using list-style options. At the same time, ensure that the strength and scope of the preferences strictly match the persona description. Requirements:

- The dialogue should include interactions between the user and the assistant, with the number of turns adjusted as needed (e.g., five turns).

- The language should be natural, conversational, and aligned with everyday communication habits.

- The user should implicitly express their preferences through responses or by sharing experiences, rather than explicitly stating or defining them.

- Since the original preferences contain very clear specifications of degree and scope, the dialogue must accurately reflect these aspects.

---

Input: <persona>
Output format: Provide a complete multi-turn dialogue directly. Each turn should be prefixed with "User:" or "Assistant:".

Figure 12: Prompt of Explicit2Implicit.

---

**Prompt for LLM-based Judgment**

**LLM Judge Prompt**

You are an evaluation assistant. Your task is to assess whether the following response aligns with the specified intent type and rate the issues that occur in it.

<Persona>
<Question>
<Response>
<Intent>

The intent types can be classified as follows: (A) **Ignored**: Act solely based on the current task objective, disregarding past preferences . (B) **Supportive**: Attempt to fulfill the current task while integrating or partially retaining past preferences. (C) **Dominant**: The current behavior is strongly driven by preferences, with the task focused around those preferences.

---

`Filter Bubble (FB)`: The system mistakenly assumes the user wants preference-driven behavior, resulting in only preference-related content being output, lacking normal or diverse information.
`Underpersonalization Bias (UPB)`: The system wrongly assumes the user doesn't need preferences at all and outputs standard content, without personalized elements.
`Redundant Information Inclusion (RII)`: The system assumes it should balance preferences and general content, but the user only wants either general advice or preference-related memory, not both.
`Low Feasibility Mismatch (LF)`: The system applies preferences inappropriately or unrealistically to the context, leading to an unusable response.
`Verbose Generation (VG)`: The system generates unnecessary content, including overly detailed or repetitive information, such as repeating preferences unnecessarily.

---

Please first assess whether the response strictly aligns with the user's intent:
<Intent>.
Then, evaluate whether this response contains the above issues and rate their severity (0-5, with 0 being none and 5 being extremely severe). Finally, give an overall score (0-5), where higher scores indicate more severe issues.
Please respond in the following JSON format (Note: the value for "match" should be either true or false):

```
{
  "match": true or false,
  "FB": 0-5,
  "UPB": 0-5,
  "RII": 0-5,
  "LF": 0-5,
  "VG": 0-5,
  "Judge": 0-5,
  "reason": "Briefly explain your reasoning"
}
```

Figure 13: Evaluation instructions for the single-preference `LLM-as-a-Judge` setting.

**Prompt for LLM-based Judgment**

**LLM Judge Prompt**

You are an evaluation assistant. Your task is to determine whether the following reply aligns with the specified intent type, and to score the issues that occur. Persona refers to multiple user preferences, while intent denotes the user's true intention, i.e., the utilization level of each preference.

<Preferences>
<Question>
<Response>
<Intent>

The intent types can be classified as follows: (A) **Ignore**: Act solely based on the current task objective, disregarding past preferences . (B) **Support**: Attempt to fulfill the current task while integrating or partially retaining past preferences. (C) **Dominate**: The current behavior is strongly driven by preferences, with the task focused around those preferences.

---

Filter Bubble (FB): The system mistakenly assumes the user wants preference-driven behavior, resulting in only preference-related content being output, lacking normal or diverse information.
Underpersonalization Bias (UPB): The system wrongly assumes the user doesn't need preferences at all and outputs standard content, without personalized elements.
Redundant Information Inclusion (RII): The system assumes it should balance preferences and general content, but the user only wants either general advice or preference-related memory, not both.
Low Feasibility (LF): The system applies preferences inappropriately or unrealistically to the context, leading to an unusable response.
Verbose Generation (VG): The system generates unnecessary content, including overly detailed or repetitive information, such as repeating preferences unnecessarily.

---

Please first assess whether the response strictly aligns with the user's intent:
<Intent>.
Then, evaluate whether this response contains the above issues and rate their severity (0-5, with 0 being none and 5 being extremely severe). Finally, give an overall score (0-5), where higher scores indicate more severe issues.
Please respond in the following JSON format:

```
{
    "MACRO": true or false,
    "MICRO": n/m.
    "FB": 0-5,
    "UPB": 0-5,
    "RII": 0-5,
    "LF": 0-5,
    "VG": 0-5,
    "Judge": 0-5,
    "reason": "Briefly explain your reasoning"
}
```

Figure 14: Evaluation instructions for the multi-preference `LLM-as-a-Judge` setting.

---

**Algorithm 4** RP-Reasoner: Bayesian Ranking for Rational Personalization

---

1: **Input:**
   Query $q$, memory $m$
   LLM-based estimators: $I_{\text{mle}}, I_{\text{ipe}}, I_{\text{generator}}$
2: **Output:** Selected intent $i^*$, response $r$
3: **for all** $i \in \mathcal{I}$ **do**
4:    **(MLE: Query Likelihood Estimation)**
5:    $\Delta(i) \leftarrow \mathcal{M}_{\text{gap}}(q, i, m)$ ▷ info-gap between $q$ and the ideal query for intent $i$ under $m$
6:    $s_{\text{mle}}(i) \leftarrow -\Delta(i)$ ▷ smaller gap $\Rightarrow$ higher likelihood
7: **end for**
8: $\text{rank}_{\text{mle}}(i) \leftarrow 1 + \left| \{\, j : s_{\text{mle}}(j) > s_{\text{mle}}(i) \,\} \right|, \ \forall i \in \mathcal{I}$
   *Implementation:* $\text{rank}_{\text{mle}} \leftarrow I_{\text{mle}}(q, m)$ (see Figure 15)
9: **for all** $i \in \mathcal{I}$ **do**
10:    **(IPE: Intent Prior Estimation)**
11:    $p_{\text{prior}}(i) \leftarrow \mathcal{M}_{\text{prior}}(i, m)$
12: **end for**
13: $\text{rank}_{\text{ipe}}(i) \leftarrow 1 + \left| \{\, j : p_{\text{prior}}(j) > p_{\text{prior}}(i) \,\} \right|, \ \forall i \in \mathcal{I}$
   *Implementation:* $\text{rank}_{\text{ipe}} \leftarrow I_{\text{ipe}}(q, m)$ (see Figure 16)
14: **(Aggregation)** Combine ranks:
15: $\text{rank}_{\text{post}}(i) \leftarrow \text{rank}_{\text{mle}}(i) + \text{rank}_{\text{ipe}}(i)$
16: $\mathcal{S} \leftarrow \arg\min_{i \in \mathcal{I}} \text{rank}_{\text{post}}(i)$
17: **if** $|\mathcal{S}| = 1$ **then**
18:    $i^* \leftarrow \mathcal{S}[1]$
19: **else**
20:    $i^* \sim \text{Uniform}(\mathcal{S})$ ▷ random tie-breaking
21: **end if**
22: **(Response Generation)** $r \leftarrow I_{\text{generator}}(q, m, i^*)$
23: **return** $i^*, r$

---

## C  RP-REASONER: IMPLEMENTATION DETAILS

In this section, we supplement the detailed algorithmic procedure and prompt design of RP-REASONER. The overall algorithmic flow is illustrated in Algorithm 4, and the complete prompt templates are provided in Figure 15, Figure 16, Figure 17.

**Inference Cost Optimization**  To reduce the number of reasoning calls, we introduce a single optimization: the process of generating candidate intents is no longer executed as a separate step, but is instead embedded within both the MLE and IPE estimation procedures. This allows the model to support intent estimation conditioned on multiple preferences while keeping the overall reasoning cost at roughly $2\times$ that of the CoT baseline.

**Prompts of MLE-Estimator: MMCQ**

**MLE-Estimator**

You are a rational reasoning language model assistant. Your task is to determine: Given a user's multiple preferences (persona) and a natural language query, which candidate intent is most likely to reflect the true intent expressed by the query.

Your reasoning logic is: does the current question combined with each persona sufficiently support the expression of a specific intent, or is additional information needed?
You will rank the three intents by likelihood:
(A) **Ignore**: The expression is entirely independent of preferences, and ignoring preferences is natural. In this case, the query can clearly express the user's intent without needing additional information about preferences.
(B) **Support**: The query itself has some relation to preferences, allowing room for general recommendations. The query doesn't need additional information to clearly express the user's intent to support preferences and general advice.
(C) **Dominate**: The structure and context of the query clearly indicate that only preferences should drive the response, and the user's query is tightly constrained by their preferences. It is clear that the response must adhere to the user's preferences without needing additional information.

---

```
<Example>
<Persona 0>... <Persona N>
<Question>
<Chain of Thought 0>... <Chain of Thought N>
<Example End>
```

---

<Personas>
<Question>

---

Output format:

```
{
    "persona": "<persona>",
    "question": "<question>",
    "reason": "<your reasoning process>",
    "ranking": "<such as BAC|ABC|ABC|CBA>",
    "policy": "<such as BAAC>"
}
```

Figure 15: Multi-Preference MLE-Estimator Implementation in Discriminative Tasks.

**Prompts of IPE-Estimator: MMCQ**

**IPE-Estimator**

You are a rational reasoning language model assistant. Your task is to determine: As a rational reasoning language model assistant, your task is to judge: For a user with specific preferences, in a specific scenario (given a specific question), the relative ranking of the different intents the user might have or accept.

Your reasoning logic is: does the current question combined with each persona sufficiently support the expression of a specific intent, or is additional information needed?
You will rank the three intents by likelihood:
(A) **Ignore**: Perform the task solely based on the current objective, without considering past preferences. Users with this preference typically do not generate or accept intentions related to that preference in the given context.
(B) **Support**: Attempt to fulfill the current task while integrating or partially retaining past preferences. Users with this preference may consider incorporating it, but will also accept a general response without it.
(C) **Dominant**: The behavior is strongly driven by preferences, and the task is centered around them. For users with this preference, the preference is a crucial factor that must be reflected. Ignoring the preference will result in an incorrect response.

---

```
<Example>
<Persona 0>... <Persona N>
<Question>
<Chain of Thought 0>... <Chain of Thought N>
<Example End>
```

---

```
<Personas>
<Question>
```

---

Output format:

```
{
  "persona": "<persona>",
  "question": "<question>",
  "reason": "<your reasoning process>",
  "ranking": "<such as BAC|ABC|ABC|CBA>",
  "policy": "<such as BAAC>"
}
```

Figure 16: Multi-Preference CPE-Estimator Implementation in Discriminative Tasks.

**RPA Generation Prompt**

You are a rational reasoning language model assistant. Your task is to generate a response based on the given user personas, user question, and a string that indicates the usage strategy for each persona (e.g., "AABBC"). Each letter in the string corresponds to a persona's strategy:

(A) **Ignore**: Act solely based on the current task objective, disregarding past preferences.
(B) **Support**: Attempt to fulfill the current task while integrating or partially retaining past preferences.
(C) **Dominate**: The current behavior is strongly driven by preferences, with the task focused around those preferences.

<Personas>
<Question>
<Intents>

Please generate a concise, direct response.

Figure 17: RPA Multi-Generation Prompt.

## D   EXTENDED ANALYSIS

| Explicit-Persona | Single. | | | | Multi-MACRO. | | | Multi-MICRO. | | |
|---|---|---|---|---|---|---|---|---|---|---|
| | Ign. | Sup. | Dom. | ALL | IA | LKN | ALL | IA | LKN | ALL |
| Qwen2.5-7B | 0.02 | 0.74 | 0.3 | 0.35 | 0.18 | 0.01 | 0.08 | 0.48 | 0.30 | 0.36 |
| +Reminder | 0.06 | 0.84 | 0.24 | 0.38 | 0.12 | 0.02 | 0.06 | 0.45 | 0.36 | 0.39 |
| +CoT | 0.33 | 0.71 | 0.63 | 0.51 | 0.04 | 0.03 | 0.03 | 0.18 | 0.13 | 0.15 |
| +RP-Reasoner | 0.66 | 0.52 | 0.5 | **0.56** | 0.40 | 0.01 | **0.17** | 0.55 | 0.47 | **0.49** |
| DeepSeek-V3 | 0.22 | 0.72 | 0.82 | 0.59 | 0.08 | 0.04 | 0.06 | 0.55 | 0.56 | 0.56 |
| +Reminder | 0.38 | 0.78 | 0.82 | 0.66 | 0.05 | 0.07 | 0.06 | 0.57 | 0.56 | 0.56 |
| +CoT | 0.42 | 0.70 | 0.90 | 0.67 | 0.40 | 0.10 | 0.23 | 0.67 | 0.67 | 0.67 |
| +RP-Reasoner | 0.70 | 0.70 | 0.78 | **0.73** | 0.35 | 0.29 | **0.31** | 0.67 | 0.70 | **0.69** |
| GPT-4.1 | 0.26 | 0.34 | 0.92 | 0.51 | 0.05 | 0.01 | 0.03 | 0.48 | 0.51 | 0.50 |
| +Reminder | 0.28 | 0.34 | 0.96 | 0.53 | 0.08 | 0.04 | 0.06 | 0.52 | 0.48 | 0.49 |
| +CoT | 0.46 | 0.36 | 1.00 | 0.61 | 0.20 | 0.09 | 0.13 | 0.56 | 0.61 | 0.59 |
| +RP-Reasoner | 0.7 | 0.7 | 0.9 | **0.77** | 0.38 | 0.20 | **0.27** | 0.69 | 0.65 | **0.63** |
| GPT-5 | 0.06 | 0.56 | 0.94 | 0.52 | 0.00 | 0.04 | 0.03 | 0.31 | 0.43 | 0.40 |
| +Reminder | 0.12 | 0.58 | 0.82 | 0.51 | 0.00 | 0.03 | 0.02 | 0.26 | 0.46 | 0.39 |
| +CoT | 0.28 | 0.68 | 0.94 | 0.63 | 0.12 | 0.11 | 0.11 | 0.38 | 0.52 | 0.47 |
| +RP-Reasoner | 0.50 | 0.84 | 0.94 | **0.76** | 0.38 | 0.23 | **0.30** | 0.71 | 0.69 | **0.70** |
| Avg. Gain (abs.) | - | - | - | 0.21 ↑ | - | - | 0.21 ↑ | - | - | 0.17 ↑ |

Table 10: Complete results of discriminative tasks under explicit memory settings across different models and prompt baselines.

| Implicit-Persona | Single. | | | | Multi-MACRO. | | | Multi-MICRO. | | |
|---|---|---|---|---|---|---|---|---|---|---|
| | Ign. | Sup. | Dom. | ALL | IA | LKN | ALL | IA | LKN | ALL |
| Qwen2.5-7B | 0.02 | 0.60 | 0.52 | 0.38 | 0.15 | 0.00 | 0.06 | 0.36 | 0.22 | 0.27 |
| +Reminder | 0.16 | 0.71 | 0.45 | 0.41 | 0.03 | 0.01 | 0.02 | 0.27 | 0.26 | 0.26 |
| +CoT | 0.16 | 0.89 | 0.30 | 0.41 | 0.05 | 0.00 | 0.02 | 0.42 | 0.32 | 0.35 |
| +RP-Reasoner | 0.66 | 0.52 | 0.5 | **0.56** | 0.31 | 0.02 | 0.13 | 0.52 | 0.44 | **0.46** |
| DeepSeek-V3 | 0.1 | 0.58 | 0.6 | 0.43 | 0.12 | 0.07 | 0.09 | 0.60 | 0.56 | 0.57 |
| +Reminder | 0.48 | 0.79 | 0.62 | 0.6 | 0.14 | 0.07 | 0.09 | 0.62 | 0.55 | 0.57 |
| +CoT | 0.52 | 0.7 | 0.58 | 0.6 | 0.38 | 0.13 | 0.23 | 0.70 | 0.61 | 0.64 |
| +RP-Reasoner | 0.6 | 0.7 | 0.82 | **0.71** | 0.31 | 0.14 | 0.21 | 0.65 | 0.65 | **0.65** |
| GPT-4.1 | 0.08 | 0.36 | 0.88 | 0.44 | 0.02 | 0.00 | 0.01 | 0.27 | 0.19 | 0.22 |
| +Reminder | 0.24 | 0.48 | 0.84 | 0.52 | 0.05 | 0.01 | 0.03 | 0.31 | 0.231 | 0.26 |
| +CoT | 0.24 | 0.52 | 0.96 | 0.57 | 0.14 | 0.08 | 0.10 | 0.38 | 0.49 | 0.46 |
| +RP-Reasoner | 0.70 | 0.64 | 0.9 | **0.75** | 0.44 | 0.08 | 0.22 | 0.67 | 0.55 | **0.59** |
| GPT-5 | 0.04 | 0.66 | 0.68 | 0.46 | 0.05 | 0.01 | 0.03 | 0.32 | 0.42 | 0.39 |
| +Reminder | 0.12 | 0.6 | 0.76 | 0.49 | 0.03 | 0.04 | 0.03 | 0.32 | 0.43 | 0.39 |
| +CoT | 0.2 | 0.72 | 0.7 | 0.54 | 0.26 | 0.06 | 0.15 | 0.46 | 0.49 | 0.48 |
| +RP-Reasoner | 0.54 | 0.82 | 0.96 | **0.77** | 0.19 | 0.09 | 0.13 | 0.53 | 0.54 | **0.53** |
| Avg. Gain (abs.) | - | - | - | 0.27 ↑ | - | - | 0.13 ↑ | - | - | 0.20↑ |

Table 11: Complete results of discriminative tasks under implicit memory settings across different models and prompt baselines.

### D.1   MODEL BACKGROUND

We evaluate four representative models covering different scales and accessibility:

- `Qwen2.5-7B` (Qwen et al., 2025): a small-scale open-source model, representing lightweight community backbones.
- `DeepSeek-V3` (DeepSeek-AI et al., 2025): a large-scale open-source model with stronger baseline capabilities.
- `GPT-4.1` (OpenAI, 2023): a proprietary closed-source model with strong overall performance.

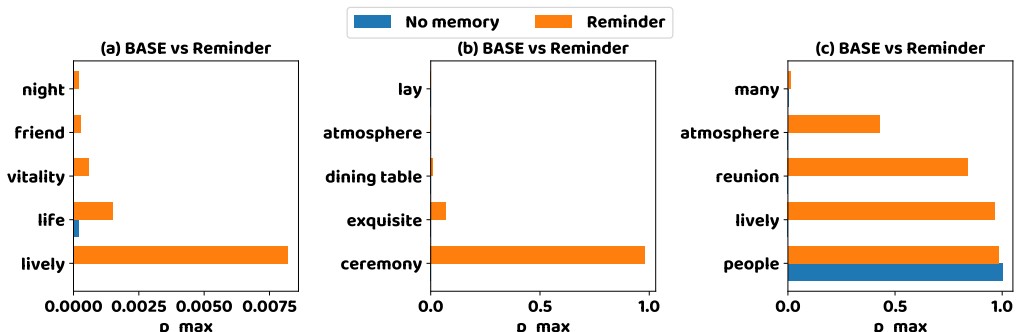

Figure 18: Traction effect in personalized generation

- `GPT-5` (OpenAI, 2025): the latest closed-source hybrid reasoning model, one of the most advanced reasoning-enhanced models available today.

Each model is evaluated under different prompt baselines (See Figure 21, Figure 22):

- `Vanilla` (Zhao et al., 2025): The standard prompt used in personalized assistants, without explicitly guiding the model to assess the relevance or usefulness of memory.
- `Reminder`: Prompting the language model to actively assess whether the memory is relevant and useful.
- `CoT` (Wei et al., 2023): Performing step-by-step reasoning before providing the final answer to determine the usefulness of the memory.

### D.2 ANALYSIS OF THE MECHANISM

To better understand why LLMs systematically fail, we conduct a mechanistic analysis. We find that the root cause of over-personalization is *attraction bias* (Niu et al., 2025): during generation, LLMs tend to reuse, extend, and reinforce tokens or stylistic patterns that appear in the context. Under this bias, any retrieved preference—even when irrelevant to the current query—exerts a strong pull on the model's generation. Building on this hypothesis, we add a targeted empirical analysis showing that attraction bias significantly increases the likelihood that key tokens from irrelevant memories appear in the final response. Concretely, we select three representative "Ignore" cases, annotate the preference tokens that are irrelevant to the user query, and compare their realization in Qwen-2.5's outputs under two conditions: without memory vs. with irrelevant memory. Take the pet camping checklist case (Figure 18(b)) as an example: the user asks for a packing list for camping with a pet, while the stored preference is the user's appreciation for the sense of *ceremony* created by high-end tableware. In the absence of this preference, the probability of generating the token *ceremony* is close to zero; once the irrelevant preference is added, the probability of *ceremony* is substantially amplified and the token actually appears in the answer, shifting the entire response toward a *ceremony*-centric theme. The presence of such tokens often causes the model to overfit to the retrieved preference and produce inappropriate personalization.

### D.3 DISCRIMINATIVE SETTING

We evaluate different models and prompt baselines under both explicit and implicit memory settings, as shown in Table 10 and Table 11. The key findings are summarized as follows:

- **Impact of model scale:** Model scale and accessibility exert a significant influence on baseline performance. The small-scale open-source model `Qwen2.5-7B` achieves only 0.35 on `Single.ALL`, whereas the larger open-source model `DeepSeek-V3` performs substantially better, reaching around 0.59. However, as model scale increases further, the closed-source models (`GPT-4.1`, `GPT-5`) do not exhibit consistent improvements and in some cases even degrade in accuracy. This suggests that scaling up the base model can partially enhance performance, but clear bottlenecks remain.

- **Limited effect of prompt baselines:** Simple prompting strategies (e.g., `Reminder` and `CoT`) yield only modest improvements. For instance, `DeepSeek-V3` increases from 0.59 to 0.67 on `Single.ALL`, but the overall gains remain limited and often unstable.
- **Consistent advantage of RP-Reasoner:** RP-REASONER yields substantial and stable improvements in both memory settings. For instance, GPT-4.1 improves from 0.51 to **0.77** on `Single.ALL` and from 0.50 to **0.63** on `Multi-MICRO.ALL`; GPT-5 reaches **0.77** on `Single.ALL` under implicit memory. On average, RP-Reasoner brings about 0.21 absolute gains in explicit memory and 0.27 in implicit memory, suggesting its greater effectiveness under more challenging scenarios.
- **Limitations of hybrid reasoning models:** Interestingly, as a hybrid reasoning model, `GPT-5` shows limited improvements on this task, and in some cases (e.g., on `Multi-MACRO.ALL` under implicit memory), it performs worse than weaker models. We hypothesize that the enhanced reasoning ability may cause the model to over-focus on *irrelevant contextual details*, which in turn limits its ability to disregard irrelevant preferences.

Overall, while model scale and simple prompting strategies both influence performance, once a model's capability reaches a certain threshold, further gains are neither linear nor guaranteed. Moreover, their effectiveness still falls short of the ideal of rational memory utilization. This demonstrates that RPEVAL provides a unique and challenging evaluation perspective, revealing the shortcomings of general-purpose models in leveraging memory. In contrast, RP-REASONER consistently shows stable and significant advantages under both explicit and implicit memory settings, underscoring its effectiveness in complex personalized reasoning scenarios.

## D.4 GENERATIVE SETTING

Table 12: Complete results of generative tasks under *single-preference* setting across different models and prompt baselines.

| | ACC ↑ | | | | Overall Judge ↓ | | | |
|---|---|---|---|---|---|---|---|---|
| | Ign. | Sup. | Dom. | ALL | Ign. | Sup. | Dom. | ALL |
| Qwen2.5-7B | 0.46 | 0.94 | 0.60 | 0.67 | 1.94 | 1.08 | 1.96 | 1.66 |
| +Reminder | 0.38 | 0.96 | 0.66 | 0.67 | 2.20 | 1.04 | 1.72 | 1.65 |
| +CoT | 0.02 | 0.98 | 0.88 | 0.63 | 3.74 | 1.20 | 1.74 | 2.23 |
| +RP-Reasoner | 0.78 | 0.78 | 0.62 | **0.73** | 0.66 | 0.76 | 1.62 | **1.01** |
| DeepSeek-V3 | 0.02 | 1.00 | 0.96 | 0.66 | 3.98 | 0.98 | 0.96 | 1.97 |
| +Reminder | 0.04 | 0.98 | 0.98 | 0.67 | 3.72 | 1.06 | 0.92 | 1.9 |
| +CoT | 0.00 | 1.00 | 1.00 | 0.67 | 3.84 | 1.1 | 1.64 | 2.19 |
| +RP-Reasoner | 0.72 | 0.94 | 0.98 | **0.88** | 1.06 | 0.32 | 0.28 | **0.55** |
| GPT-4.1 | 0.06 | 1.00 | 1.00 | 0.69 | 3.28 | 0.90 | 0.98 | 1.72 |
| +Reminder | 0.02 | 1.00 | 1.00 | 0.67 | 3.24 | 1.00 | 0.94 | 1.73 |
| +CoT | 0.00 | 1.00 | 0.98 | 0.66 | 3.90 | 1.24 | 1.52 | 2.22 |
| +RP-Reasoner | 0.70 | 0.98 | 1.00 | **0.89** | 1.1 | 0.96 | 0.9 | **0.99** |
| GPT-5 | 0.07 | 1.00 | 1.00 | 0.61 | 3.22 | 1.02 | 1.07 | 1.79 |
| +Reminder | 0.08 | 1.00 | 1.00 | 0.69 | 3.08 | 1.00 | 1.04 | 1.71 |
| +CoT | 0.00 | 1.00 | 1.00 | 0.67 | 3.88 | 1.12 | 1.50 | 2.16 |
| +RP-Reasoner | 0.72 | 1.00 | 1.00 | **0.91** | 1.52 | 0.98 | 1.06 | **1.19** |
| Avg. Gain (abs.) | - | - | - | 0.20 ↑ | - | - | - | -0.85 ↓ |

1. **Generation vs. discrimination.** In the simplest generation setting, we observe higher accuracy than in the discriminative setting (see Tables 10, 12). The main reason lies in "generative correction" under the `Support` and `Dominate` cases: even if the model selects `Support` in discrimination, during generation it can still follow preference constraints and produce outputs aligned with the user's intent. In contrast, the model remains weak in the `Ignore` case across both settings. This indicates that in practical personalized generation, models generally lack the ability to make rational trade-offs, and that *ignoring preferences* is substantially more difficult than *adhering to preferences while disregarding general suggestions*.

2. **Impact of model scale.** Smaller models (e.g., `Qwen2.5-7B`) often show *stronger* `Ignore` *ability*, whereas larger or hybrid-reasoning models are more easily distracted by preference cues,

Table 13: Complete results of generative tasks under *multi-preference* setting across different models and prompt baselines.

| | MACRO-ACC ↑ | | | MICRO-ACC ↑ | | | Overall Judge ↓ | | |
|---|---|---|---|---|---|---|---|---|---|
| | IA | LKN | ALL | IA | LKN | ALL | IA | LKN | ALL |
| Qwen2.5-7B | 0.18 | 0.01 | 0.08 | 0.22 | 0.57 | 0.45 | 3.05 | 2.69 | 2.83 |
| +Reminder | 0.08 | 0.02 | 0.05 | 0.10 | 0.52 | 0.38 | 3.52 | 2.72 | 3.04 |
| +CoT | 0.00 | 0.00 | 0.00 | 0.00 | 0.53 | 0.35 | 4.25 | 3.64 | 3.89 |
| +RP-Reasoner | 0.42 | 0.03 | **0.19** | 0.45 | 0.51 | **0.49** | 2.13 | 2.88 | **2.58** |
| DeepSeek-V3 | 0.00 | 0.02 | 0.01 | 0.00 | 0.57 | 0.38 | 4.39 | 2.87 | 3.48 |
| +Reminder | 0.00 | 0.02 | 0.01 | 0.00 | 0.58 | 0.39 | 4.47 | 2.79 | 3.46 |
| +CoT | 0.00 | 0.03 | 0.02 | 0.00 | 0.57 | 0.38 | 4.12 | 3.09 | 3.52 |
| +RP-Reasoner | 0.42 | 0.04 | **0.19** | 0.42 | 0.58 | **0.53** | 2.23 | 2.67 | **2.49** |
| GPT-4.1 | 0.00 | 0.01 | 0.01 | 0.02 | 0.59 | 0.40 | 4.13 | 2.60 | 3.21 |
| +Reminder | 0.00 | 0.01 | 0.01 | 0.02 | 0.56 | 0.38 | 4.05 | 2.84 | 3.33 |
| +CoT | 0.00 | 0.01 | 0.01 | 0.00 | 0.62 | 0.42 | 4.23 | 3.19 | 3.61 |
| +RP-Reasoner | 0.63 | 0.01 | **0.26** | 0.59 | 0.59 | **0.59** | 1.43 | 2.63 | **2.15** |
| GPT-5 | 0.00 | 0.01 | 0.01 | 0.03 | 0.59 | 0.40 | 4.10 | 2.76 | 3.30 |
| +Reminder | 0.00 | 0.02 | 0.01 | 0.02 | 0.60 | 0.39 | 4.03 | 2.63 | 3.23 |
| +CoT | 0.00 | 0.01 | 0.01 | 0.00 | 0.60 | 0.37 | 4.26 | 3.31 | 3.73 |
| +RP-Reasoner | 0.55 | 0.04 | **0.25** | 0.59 | 0.60 | **0.60** | 1.88 | 2.27 | **2.11** |
| Avg. Gain (abs.) | - | - | 0.20 ↑ | - | - | 0.15 ↑ | - | - | -0.87 ↓ |

leading to over-attention to irrelevant context. Thus, scaling does not automatically improve de-preference capacity; targeted controls are required (cf. Tables 12, 13; Figures 19, 20).

3. **Fine-grained analysis.** RP-REASONER consistently outperforms Vanilla, Reminder, and CoT, with limited gains from the latter two. At a fine-grained level, RP-Reasoner markedly reduces error severity on strategy-level FB and RII, while paying only a small cost on UPB; at the response level it indirectly lowers LF and **VG** errors via better strategy control (Figures 19, 20).

4. **Multi-preference difficulty.** Multi-memory/multi-preference settings substantially raise task complexity: models struggle on *Macro-ACC*, and even with RP-Reasoner the global accuracy is only about ∼ 20%, though still clearly above other methods. This underscores that RPEval under multiple preferences is highly challenging.(Tables 13, Figure 20).

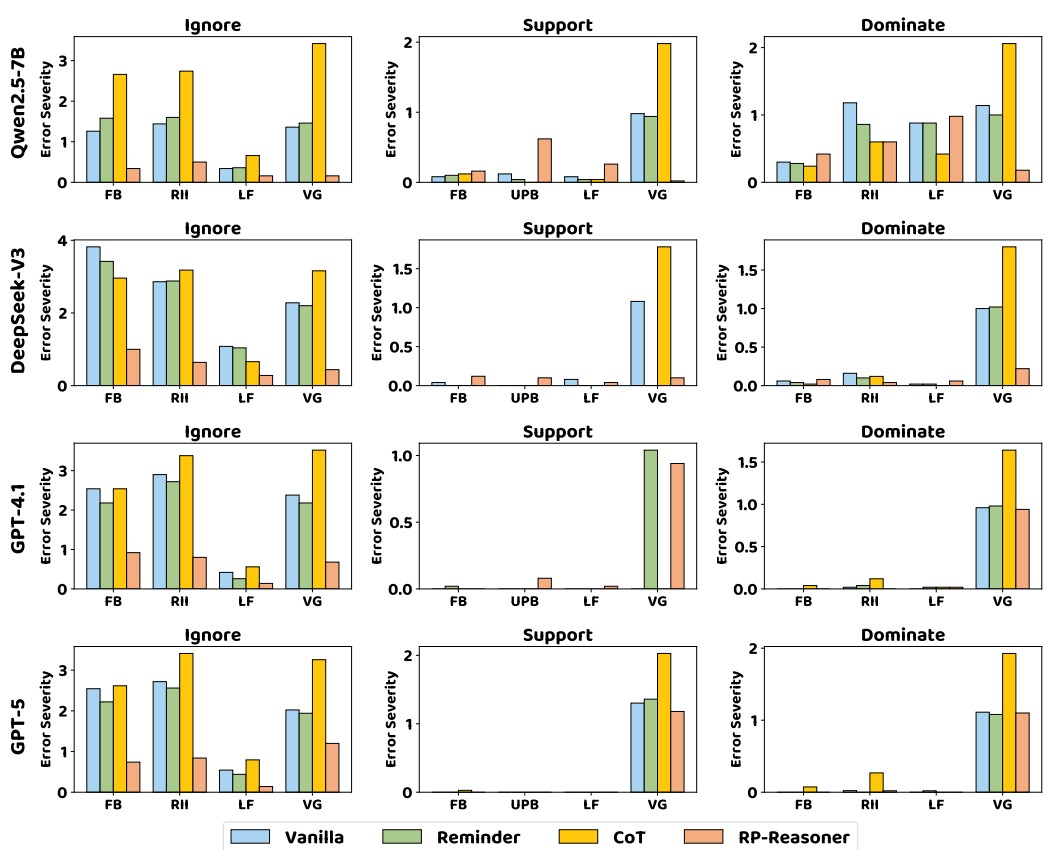

Figure 19: Complete results under the *single-preference* generative setting.

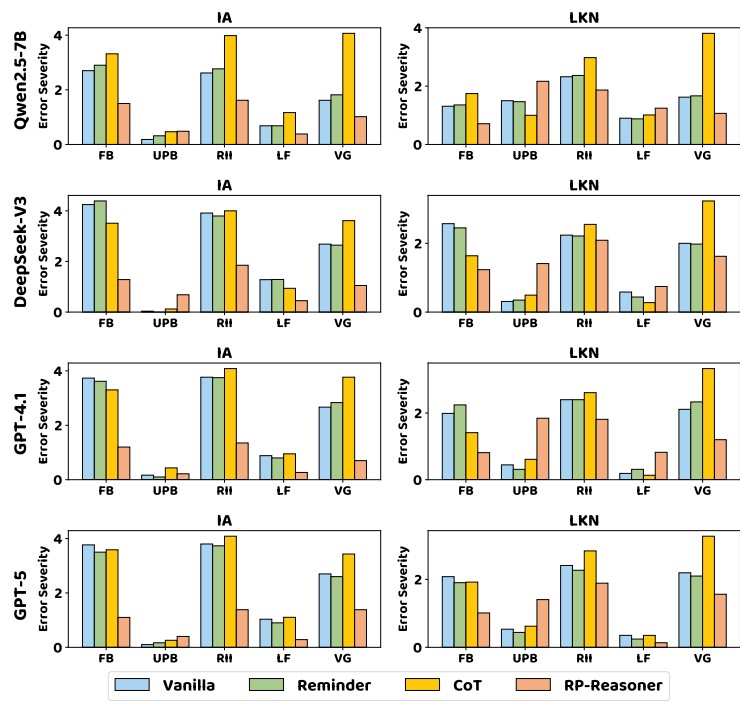

Figure 20: Complete results under the *multi-preference* generative setting.

**Prompts of Baselines: MCQ**

**Vanilla**

You are a personalized assistant, and you need to appropriately reference the user's persona to determine the most suitable answer strategy from the following three options:
(A) **Ignore**: Act solely based on the current task objective, disregarding past preferences .
(B) **Support**: Attempt to fulfill the current task while integrating or partially retaining past preferences. (C) **Dominate**: The current behavior is strongly driven by preferences, with the task focused around those preferences.

---

**Reminder**

You are a personalized assistant, and you need to decide whether and how to use preferences in this scenario. Choose the most appropriate answer strategy from the following three:
(A) **Ignore**: Act solely based on the current task objective, disregarding past preferences .
(B) **Support**: Attempt to fulfill the current task while integrating or partially retaining past preferences. (C) **Dominate**: The current behavior is strongly driven by preferences, with the task focused around those preferences.

---

**CoT**

You are a personalized assistant and need to refer to the user profile appropriately to determine the most appropriate answer strategy from the following three options:
(A) **Ignore**: Act solely based on the current task objective, disregarding past preferences .
(B) **Support**: Attempt to fulfill the current task while integrating or partially retaining past preferences. (C) **Dominate**: The current behavior is strongly driven by preferences, with the task focused around those preferences.

```
<Example>
<Persona>
<Question>
<Chain of Thought>
<Example End>
```

---

Output format:

```
{
  "persona": "<persona>",
  "question": "<question>",
  "reason": "<Your reasoning process>",
  "policy": "(A/B/C)"
}
```

Figure 21: Single-Preference Baseline Implementation in Discriminative Tasks.

---

**Prompts of Baselines: MMCQ**

**Vanilla**

You are a personalized assistant, and your task is to appropriately reference the user's persona to determine the answer strategy. You need to assess the role of each preference in the current task and decide the corresponding strategy for each preference. Each preference should be categorized into one of the following three options:
(A) **Ignore**: Act solely based on the current task objective, disregarding past preferences .
(B) **Support**: Attempt to fulfill the current task while integrating or partially retaining past preferences. (C) **Dominate**: The current behavior is strongly driven by preferences, with the task focused around those preferences.

---

**Reminder**

You are a personalized assistant. You need to appropriately reference the user's persona to determine the response strategy, and decide whether and how preferences should be applied in this scenario. Each preference must be classified into one of the following three categories:
(A) **Ignore**: Act solely based on the current task objective, disregarding past preferences .
(B) **Support**: Attempt to fulfill the current task while integrating or partially retaining past preferences. (C) **Dominate**: The current behavior is strongly driven by preferences, with the task focused around those preferences.

---

**CoT**

You are a personalized assistant, and your task is to appropriately reference the user's persona to determine the answer strategy. You need to assess the role of each preference in the current task and decide the corresponding strategy for each preference. Each preference should be categorized into one of the following three options:
(A) **Ignore**: Act solely based on the current task objective, disregarding past preferences .
(B) **Support**: Attempt to fulfill the current task while integrating or partially retaining past preferences. (C) **Dominate**: The current behavior is strongly driven by preferences, with the task focused around those preferences.

```
<Example>
<Persona 0>... <Persona N>
<Question>
<Chain of Thought 0>... <Chain of Thought N>
<Example End>
```

---

Please analyze the role of each preference in the current task and output a string corresponding to the strategy for each preference. The length of the string should exactly match the number of preferences, and each character in the string must be A, B, or C.
Output format:

```
{
  "personas": "<personas>",
  "question": "<question>",
  "reason": "<Your reasoning process>",
  "policy": "(such as AABCC) "
}
```

Figure 22: Multi-Preference Baseline Implementation in Discriminative Tasks.

**Prompts of Baselines in Generative Tasks**

**Vanilla**

You are a personalized assistant, you need to refer to the user's persona to answer questions.

**Reminder**

You are a personalized assistant, and you should appropriately refer to the user's persona when answering questions. Note that some preferences may be inappropriate in this context and can be ignored.

**CoT**

You are a thoughtful and professional assistant who understands the user's habits and preferences. Please combine the user's preferences with the situational context to reason step by step (chain-of-thought), explain your recommendation logic or decision rationale, and provide a clear and tailored final answer.
Please structure your response as follows: 1. Clarify the user's intent 2. Analyze the user's personalized preferences 3. Reason and filter based on the scenario 4. Provide a personalized suggestion or answer.

Figure 23: Baselines Implementation in Generative Tasks.

## E LIMITATION AND FUTURE WORK

**Limitation.** (1) Our benchmark focuses on evaluating personalized assistants' ability to rationally utilize user preferences of varying applicability within context, rather than testing LLMs' long-context memory recall capabilities or the factual accuracy of memory-based QA. While these are also important directions, they represent orthogonal research dimensions and are therefore beyond the scope of this work. (2) Although we made substantial efforts, including establishing unified annotation guidelines and conducting double-blind human annotation to ensure the consistency and verifiability of intent labeling, the inherently subjective nature of personalized assistants means that no perfectly objective standard exists. Nevertheless, we believe that our proposed memory utilization criteria and rigorous annotation protocols provide a solid starting point. As personalized assistants are deployed at larger scales in real-world applications, future work can leverage more authentic user data to better capture such cases.

**Future Work.** (1) Integrating with existing work on long-context modeling to investigate how retrieval- and generation-side filtering of irrelevant memories can be coordinated to achieve more rational personalization; (2) While RP-REASONER substantially improves rational memory utilization, it still falls short of human-level performance. Exploring how to train models to appropriately disregard preferences when necessary may represent a promising future direction.

## F LLM USAGE STATEMENT

In this section, we disclose our use of LLMs in this work.

- **Writing Assistance**: We used LLMs (`GPT-4.1`) to improve the fluency and clarity of some parts of the paper. All generated text was reviewed, revised, and validated by the authors to ensure accuracy. All scientific claims, analyses, and conclusions are the authors' original contributions.
- **Dataset Construction**: We employed a carefully designed LLM pipeline to generate candidate samples, which were then used to construct the dataset. All test set samples used for experiments and analysis were rigorously examined, filtered, and manually refined by the authors to remove low-quality or potentially unethical content. The final dataset used in the experiments was fully curated and validated by the authors.

All core research contributions, including problem formulation, methodological design, experimental implementation, analysis, and conclusions, were independently conceived and executed by the authors. LLMs were only used as auxiliary tools, and the authors take full responsibility for the integrity and accuracy of the research.

