# OpenReview forum: "Benchmarking and Enhancing Rational Preference Utilization for Personalized Assistants: A Pragmatic View"
_ICLR.cc/2026/Conference — ICLR 2026 Conference Withdrawn Submission_

### Official Review · Reviewer_Gvb6 · 2025-10-17

**Soundness:** 2
**Presentation:** 2
**Contribution:** 2
**Rating:** 2
**Confidence:** 4

**Summary:**

This work addresses the question of how LLMs should incorporate user-specific memories, arguing that naively applying personalization can lead to poor outputs.  To study this problem, the authors introduce a synthetic benchmark of underspecified user queries labeled according to intent.  The dataset is constructed through multi-stage LLM generation, and evaluated primarily using an LLM judge.  To help the LLM address this query underspecification problem, the authors proposed a method, RP-Reasoner, a heuristic prompting scheme inspired by Rational Personalization Acts that combines a query likelihood term with an intent prior to decide the appropriate personalization level based on the inferred user intent.  Experiments suggest that this method outperforms simpler prompting baselines.

**Strengths:**

The paper is well-motivated and timely.  LLM personalization is an area of great interest to the community, and the authors rightly point out that existing approaches are largely naive, and better algorithms are needed for true contextual personalization.

**Weaknesses:**

My major concern is that the paper is organized around the idea of RPAs, but I am not sure what new insights are gained from taking this viewpoint.  The core claim, that personalized systems should not blindly apply stored preferences, but instead should infer intent and use context to decide whether/what to personalize, is a long-standing theme in recommender systems and LLM personalization.  For example, here is a survey paper on context-aware recommender systems from 2011 with over 3K citations: https://ojs.aaai.org/aimagazine/index.php/aimagazine/article/view/2364.  I think L2 personalization is always taken for granted as the goal.

Also, I find that the description of “memory utilization” does not really align with the task or proposed method; this is more about properly conditioning on a persona.  There are many interesting questions around how to use a constantly evolving memory store of past user interactions (from this and other users), but this work does not grapple with that.

I was excited when I read the first sentence of the motivation, that “This work centers on the duality of personalization, particularly the potential risks.”  However, I don’t think the issues addressed in this paper are truly risks, they’re just cases of bad personalization.  When I think of the risks of personalization, I think of addiction, sycophancy, and a range of other unhealthy feedback loops and phenomena.  It would have been interesting to see the paper focus on some of these real risks.

I am unconvinced that the LLM judge has been thoroughly validated.  Figure 3c is underexplained, and from what I can tell agreement levels are not that high.  What check or significance test is done here?  I looked at examples in the appendix, and I actually disagreed with the filter bubble example (why can’t the parent want the child to have a small serving of protein with their vegetables?), so I am worried about how well evaluation might function here.

I am also unconvinced that baselines are very competitive.  How was the CoT prompt optimized?  It seems like a carefully prompted reasoning model should be competitive at this.  Overall, it is hard to draw strong conclusions from the experiments; while the proposed method performs best, it is somewhat unsurprising given that it is bespoke to the unique benchmark created by the authors.  I am not sure of the broad applicability of this method.

**Questions:**

- What significance testing was done w.r.t. agreement between LLM judge and human annotions?
 - How was the CoT prompt optimized?

---

> ### Author Response · Authors · 2025-11-22
> **Rebuttal by Authors [W1, W2]**
>
> Thank you for your insightful and helpful feedback. We have conducted several supplementary experiments and gathered additional materials; (provided separately), and summarize the key findings here. We hope these responses adequately address your points and would be grateful if you would consider increasing your score.
>
> ---
>
> **Response to Weakness 1:  No new insight from RPAs**
>
> We thank the reviewer for raising this important point. We fully agree that “inferring user intent and conditioning on context before deciding whether and how to personalize” (what we term L2-level personalization) has long been a widely acknowledged goal in the literature on context-aware recommendation. Our work does not claim this idea itself to be new. Rather, we argue that in the concrete setting of **LLM-based Personalized Assistant**, **this perspective has not yet been systematically formalized, evaluated, and operationalized.**
>
> 1. **L2-level personalization has not been modeled or evaluated in LLM personalization.** To the best of our knowledge, existing work on LLM memory and personalization almost universally assumes that “whenever relevant information is present in memory, it should be used,” which corresponds to what we define as L1-level personalization, without distinguishing when personalization is beneficial, redundant, or even harmful. Through the RPA framework and the RPEval benchmark, we are **the first to systematically characterize and evaluate the problem of rational personalization in LLM assistants, and to turn it into a measurable evaluation task**.
>
> 2. **LLM personalization differs fundamentally from traditional context-aware recommendation in data and decision structure.** In LLM assistants, personalization primarily relies on a small number of sparse, free-form textual memories, rather than large-scale, structured interaction logs; at the same time, the action space is open-ended natural language and tool use, rather than ranking over a fixed item set. Under this setting, conventional context-aware recommendation datasets, evaluation protocols, and learning paradigms based on large-scale exposure/click logs are not directly applicable. To investigate this long-standing theme of personalization in the context of LLM-based PA, there is an urgent need for new datasets and methodological advances.
> 3. **RPA further yields a practical pragmatic reasoning framework.** Building on pragmatics and counterfactual reasoning, RPA not only reveals systematic over-personalization in current LLM personalization practices, but also leads to the concrete **RP-Reasoner** framework. Without relying on large-scale user–item logs, RP-Reasoner demonstrates substantially more rational personalization behavior on both the rpeval benchmark and real-world data.
>
> ---
>
> **Response to Weakness 2:  Misaligned memory framing**
>
> We thank the reviewer for the comment.  We acknowledge that leveraging continuously evolving historical interaction memories from the current user and from other users is an important and promising research direction. However, regardless of how a memory system is designed, there is ultimately a stage where the **LLM must condition its generation on both the current query and the user preference information**.
>
> In the setting of LLM-based personalized assistants, there are two key characteristics: (1) the available user memories are often sparse and partial; and (2) user requests are highly open-ended and diverse. Thus, the content retrieved by the memory module will **inevitably introduce a certain amount of irrelevant or secondary information into the LLM’s context**. This practical consideration motivates us to focus on the stage of **memory utilization**, namely how the LLM should use a set of memories already selected by the memory system during generation in order to produce more appropriate and rationally personalized outputs.

---

> ### Author Response · Authors · 2025-11-22
> **Rebuttal by Authors [W3, W4 & Q1]**
>
> **Response to Weakness 3:  Ignores deeper harmful loops**
>
> We thank the reviewer for the thoughtful comments. In this paper, we use the term *“risk”* to refer primarily to **system-side personalization risks**—that is, cases where personalized information misleads the model and causes its responses to deviate from the user’s true intent. Along this dimension, we identify and empirically distinguish five typical categories of system-side error patterns. **For real-world commercial AI assistants, these widespread and frequent forms of “bad personalization” already have a substantial negative impact on user experience, and therefore constitute a practical and non-trivial source of risk.**
>
> We agree that addiction, sycophancy, and various unhealthy feedback loops are important behavioral risks. In practice, however, such phenomena are typically driven by multiple interacting factors, and their causal relationship with personalization alone is difficult to isolate. In contrast, the risks studied in this work are directly observable and can be clearly attributed to the personalization mechanism itself. A careful characterization and mitigation of our system-side risks is a necessary prerequisite for understanding and preventing more complex behavioral risks.
>
> ---
>
> **Response to Weakness 4 & Question 1:  LLM judge undervalidated**
>
> **Response:** We thank the reviewer for raising this concern. We have clarified the validation of the LLM-as-judge setup in the revised version.
>
> **1. Validation of LLM-as-judge.** We ask human annotators to follow **exactly the same evaluation instructions** as the LLM judge, i.e., to assign an ordinal score from 0–5 for each of the five personalization error types. Figure 3(c) reports the agreement between the LLM judge and human annotations on these 0–5 ordinal scores, measured by **quadratic weighted Cohen’s kappa (QWK)**, a standard statistic for ordered labels. The overall agreement is QWK = 0.87, which is typically interpreted as **a high agreement** in the literature on inter-rater reliability [1].
>
> **2. On the “filter bubble” example.** We appreciate the reviewer’s careful reading of the examples. Our use of the term *filter bubble* follows standard definitions in the existing literature [2], where it is typically defined as over-homogenization of a user’s information exposure induced by personalization, rather than a judgment about the feasibility of any single recommendation. In the cited case, the assistant repeatedly over-amplifies a weakly relevant signal (the parent’s own fitness preference) and effectively compresses the candidate space into “vegetable recipes with added protein”, with **other reasonable meal plans being rarely or never surfaced**. This pattern—personalization-induced narrowing of the exposed option set—is what we label as a filter-bubble error.

---

> ### Author Response · Authors · 2025-11-22
> **Rebuttal by Authors [W5 & Q2]**
>
> **Response to Weakness 5 & Question 2:  Baselines not competitive enough**
>
> **1. Design of the CoT prompt.** We provide the full CoT prompt in the in the detailed document). In brief, we (a) explicitly instruct the model when to choose each type of preference utilization strategy, and (b) provide three carefully constructed, step-by-step in-context examples for each utilization mode. We will clarify this design in the main text and point readers more clearly to the full prompt in the appendix.
>
> **2. Additional reasoning-based baselines.**
>
> To further address this concern, we have added two stronger reasoning baselines: **self-consistency (SC)** [3] and **self-refine** [4]. Both methods build directly on the carefully designed CoT prompt described above, but employ multiple sampled reasoning paths (SC) or iterative refinement (self-refine) to improve robustness. These baselines are therefore strictly stronger than the single-pass CoT baseline, but also incur higher inference cost than RP-Reasoner. In our updated experiments, SC and self-refine do improve over plain CoT on some metrics, but still consistently underperform RP-Reasoner, while using substantially more model calls.
>
> | Method            | MACRO-IA | MACRO-LKN | MACRO-ALL | MICRO-IA | MICRO-LKN | MICRO-ALL |
> | :---------------- | -------- | --------- | --------- | -------- | --------- | --------- |
> | GPT-4.1           | 0.05     | 0.01      | 0.03      | 0.48     | 0.51      | 0.50      |
> | + CoT-SC [3]      | 0.22     | 0.08      | 0.13      | 0.56     | 0.61      | 0.59      |
> | + Self-refine [4] | 0.18     | 0.07      | 0.11      | 0.57     | 0.56      | 0.57      |
> | + RP-Reasoner     | **0.38** | **0.20**  | **0.27**  | **0.69** | **0.65**  | **0.63**  |
> | GPT-5             | 0.00     | 0.04      | 0.03      | 0.31     | 0.43      | 0.40      |
> | + CoT-SC [3]      | 0.17     | 0.09      | 0.12      | 0.45     | 0.55      | 0.52      |
> | + Self-refine [4] | 0.18     | 0.03      | 0.09      | 0.46     | 0.45      | 0.45      |
> | + RP-Reasoner     | **0.38** | **0.23**  | **0.30**  | **0.71** | **0.69**  | **0.70**  |
>
>
>
> [1] The measurement of observer agreement for categorical data
>
> [2] *The Filter Bubble: What the Internet Is Hiding from You*. 2011.
>
> [3] Self-Consistency Improves Chain of Thought Reasoning in Language Models.
>
> [4] Self-Refine: Iterative Refinement with Self-Feedback.
>
> ---
>
> We hope these clarifications and the additional materials detailed address your concerns. We believe RPA offers a valuable contribution to personalized assistant and respectfully request you consider these points in your final assessment.
>
> We provide a document outlining the CoT prompt here:  https://drive.google.com/file/d/1uJ_01Bqsw31cmlGAuHSGpNyJ9_QDwEc8/view?usp=sharing

---

> > ### Comment · Reviewer_Gvb6 · 2025-11-22
> >
> > I thank the authors for their detailed responses to my review.  I have no further questions at this time.
> >
> > I have raised my score, but still consider this paper to be below the threshold for acceptance.

---

> > > ### Author Response · Authors · 2025-11-24
> > >
> > > Dear reviewer Gvb6,
> > >
> > > Thanks very much for your feedback. In the rebuttal, we try our best to answer your questions one by one. If you have further questions, we are very happy to discuss more about them.
> > >
> > > We sincerely thank you for your time in reviewing our paper and our responses

---

> > > > ### Comment · Reviewer_Gvb6 · 2025-11-24
> > > >
> > > > Thank you to the authors.  I have significant remaining concerns about the work, particularly around the potential impact of the RPA framing and the quality of the data and evaluation protocol.  Also, I share many of the other reviewers’ concerns.

---

### Official Review · Reviewer_tTCm · 2025-11-02

**Soundness:** 3
**Presentation:** 2
**Contribution:** 2
**Rating:** 4
**Confidence:** 4

**Summary:**

This paper tackles over-personalization in LLM assistants—when stored preferences get applied even when they shouldn't be. The authors propose RPA (a pragmatic framework), RPEval (a benchmark with 8K samples), and RP-Reasoner (a Bayesian reasoning method). Results show current LLMs struggle badly at deciding when to ignore preferences, with RP-Reasoner bringing ~35% improvement.

**Strengths:**

1. Important problem: Over-personalization is a real issue that hasn't gotten enough attention. The sleep music example in Figure 1 perfectly illustrates why this matters.
2. Well-designed benchmark: The preference inversion strategy is clever—generating queries first, then creating preferences that should/shouldn't apply. The error taxonomy (FB, RII, UPB, etc.) is also useful for understanding failure modes.
3. Strong empirical results: 80% resolution on real commercial system bad cases is impressive and shows practical value.
4. Interesting finding about model scale: The counterintuitive result that stronger models (GPT-5) can be worse at ignoring irrelevant preferences is worth highlighting.

**Weaknesses:**

1. Heavy GPT-4 dependency: The whole dataset comes from GPT-4.1 generation. This feels circular when you're then evaluating GPT-4.1/GPT-5 on it. How do you know the benchmark doesn't just measure "how well does model X mimic GPT-4's personalization decisions"? Would've been better to ground this in real user interaction data.
2. Subjectivity issue not fully resolved: The paper acknowledges when to apply preferences is subjective, but doesn't provide inter-annotator agreement scores. Also, only ~1K samples get human annotation—what about the other 7K?
3. Still far from human performance: On Single.ALL, best result is 0.77 vs 0.95 for humans. What's causing this gap? The paper doesn't dig into what RP-Reasoner still gets wrong.

**Questions:**

1. What's the inter-annotator agreement? How were disagreements handled?
2. Computational cost: how much slower/expensive is RP-Reasoner vs baselines?
3. The "Ignore" intent is hardest for models—why? Is there something fundamental about LLMs that makes them reluctant to discard context?
4. How would this work with actual user behavior data instead of synthetic scenarios?

---

> ### Author Response · Authors · 2025-11-22
> **Rebuttal by Authors [W1, W2 & Q1, W3]**
>
> Thank you for your insightful and helpful feedback. We have conducted several supplementary experiments and gathered additional materials; (provided separately), and summarize the key findings here. We hope these responses adequately address your points and would be grateful if you would consider increasing your score.
>
> ---
> **Response to Weakness 1: GPT-4 dependency**
>
> 1. Thank you for the comment. We would like to emphasize that although the data are synthesized by GPT-4.1, the construction process is **not** simply letting GPT-4 “freely invent” a personalization logic. In particular, when  gconstruct preferences, we follow **human-designed generation guidelines** (see Table 1 in the detailed document). to constrain GPT-4’s behavior. These guidelines are never exposed to the evaluated models during testing.
>
> 2. Furthermore, even when the evaluated models are also GPT-4 or GPT-5, their relatively weak performance on personalization decisions suggests that the benchmark is **not** merely measuring “how well a model imitates GPT-4’s personalization behavior.”
>
> 3. Finally, because personalized assistants are still an emerging direction, real user interaction logs are typically sparse, noisy, and, due to privacy concerns, difficult to use for constructing large-scale, systematic, and *public* benchmarks. For these reasons, we follow prior work [1] in adopting synthetic data to obtain a controlled and reproducible evaluation setting.
>
> ---
>
> **Response to Weakness 2 & Question 1: Subjectivity issue not fully resolved**
>
> Thank you for raising this important point. We will highlight the corresponding inter-annotator agreement results more clearly in the revised version.  Below, we briefly summarize the agreement statistics and describe our procedure for resolving annotation disagreements **(see Algorithm 1 in the detailed document)**.
>
> For the full 8K dataset, we randomly sampled approximately 1,000 instances for human annotation. Each instance was independently annotated by two annotators in a blind setting . We then compared both annotators’ labels wcith the LLM-generated label. Whenever at least one annotator disagreed with the LLM label, we asked the disagreeing annotators to review the LLM’s rationale.
>
> - If all disagreeing annotators accepted that their initial annotation was incorrect, we kept the sample.
> - If **any** annotator maintained disagreement after reviewing the rationale, we discarded the sample entirely.
>
> The initial inter-annotator agreement was **91.86%**.
>
> Among the remaining **8.14%** disputed samples, we performed the above disambiguation process:
>
> - **4.37%** of samples were retained after adjudication,
> - **3.77%** were discarded due to unresolved disagreement.
>
> Given this high level of human consistency during the blind annotation stage, we consider the remaining 7K LLM-generated samples to have reasonably high confidence as well.
>
> ---
>
> **Response to Weakness 3: Still far from human performance**
>
> **1. Source of the gap.** In the *discriminative* setting, we manually inspected RP-Reasoner’s errors. We find that roughly **19.3%** of the errors are due to **over-personalization** (applying preferences when they should be ignored), while the remaining **3.3%** are due to under-personalization.
>
> **2. Error-type Analysis.** In the generative setting, we further conduct a fine-grained analysis of the residual errors made by RP-Reasoner **(see Figure 1 in the detailed document)**. The results show that RP-Reasoner significantly reduces most FB and RII issues, but some over-personalization of this kind still remains. In addition, for certain backbone models, we also observe cases of UPB.

---

> ### Author Response · Authors · 2025-11-22
> **Rebuttal by Authors [Q2, Q3, Q4]**
>
> **Response to Question 2: Computational cost**
>
> In our implementation, we optimize the RP-Reasoner pipeline to support intent estimation over multiple preferences simultaneously, keeping its inference cost at roughly **2×** that of the CoT baseline. To further assess the efficiency of RP-Reasoner, we additionally include **self-consistency** [2] and **self-refine** [3] as baselines, whose reasoning costs are comparable to or higher than RP-Reasoner. The results show that RP-Reasoner still outperforms both of these baselines.
>
> | Method            | MACRO-IA | MACRO-LKN | MACRO-ALL | MICRO-IA | MICRO-LKN | MICRO-ALL |
> | :---------------- | -------- | --------- | --------- | -------- | --------- | --------- |
> | GPT-4.1           | 0.05     | 0.01      | 0.03      | 0.48     | 0.51      | 0.50      |
> | + CoT-SC [2]      | 0.22     | 0.08      | 0.13      | 0.56     | 0.61      | 0.59      |
> | + Self-refine [3] | 0.18     | 0.07      | 0.11      | 0.57     | 0.56      | 0.57      |
> | + RP-Reasoner     | **0.38** | **0.20**  | **0.27**  | **0.69** | **0.65**  | **0.63**  |
> | GPT-5             | 0.00     | 0.04      | 0.03      | 0.31     | 0.43      | 0.40      |
> | + CoT-SC [2]      | 0.17     | 0.09      | 0.12      | 0.45     | 0.55      | 0.52      |
> | + Self-refine [3] | 0.18     | 0.03      | 0.09      | 0.46     | 0.45      | 0.45      |
> | + RP-Reasoner     | **0.38** | **0.23**  | **0.30**  | **0.71** | **0.69**  | **0.70**  |
>
> ---
>
> **Response to Question 3:  "Ignore" intent is hardest for models—why?**
>
> Thank you for pointing this out. We believe the difficulty of the Ignore intent stems from a well-documented property of large language models known as attraction bias [4]: during generation, LLMs tend to reuse extend, and reinforce tokens or stylistic patterns that appear in the context. Under this bias, any retrieved preference—even when irrelevant—exerts a pull on the model’s generation. Building on this hypothesis, we add a targeted empirical analysis showing that attraction bias significantly increases the likelihood that key tokens from irrelevant memories appear in the final response (see Figure 2 in the detailed document).
>
> ---
>
> **Response to Question 4:  How would this work with actual user behavior data instead of synthetic scenarios?**
>
> In our real-world experiments, we collaborate with an online personal-assistant product team, who provided **over-personalization failure cases** observed in production. These cases were manually converted into the same data format as rpeval and then evaluated with RP-Reasoner. We find that RP-Reasoner was able to resolve most of these cases, and the failure patterns it addressed (e.g., FB and RII) closely matched those captured by our synthetic benchmark.
>
> ----
>
> We hope these clarifications and the additional experiments and materials detailed in the full rebuttal document address your concerns. We believe RPA offers a valuable contribution to personalized assistant and respectfully request you consider these points in your final assessment.
>
> We provide a document outlining our experimental results here:  https://drive.google.com/file/d/1oIYJONLdVv2uRoePJpVpJ3a5tBJ76VdZ/view?usp=sharing
>
> [1] Do LLMs Recognize Your Preferences? Evaluating Personalized Preference Following in LLMs
>
> [2] *Self-Consistency Improves Chain of Thought Reasoning in Language Models.*
>
> [3] *Self-Refine: Iterative Refinement with Self-Feedback.*
>
> [4] Llama See, Llama Do: A Mechanistic Perspective on Contextual Entrainment and Distraction in LLMs

---

> ### Author Response · Authors · 2025-11-28
>
> Dear Reviewer,
>
> As the author-reviewer discussion period will end soon, we would appreciate it if you could check our response to your review comments. This way, if you have further questions and comments, we can still reply before the author-reviewer discussion period ends. Thank you very much for your time!

---

### Official Review · Reviewer_vGZm · 2025-11-02

**Soundness:** 3
**Presentation:** 4
**Contribution:** 3
**Rating:** 6
**Confidence:** 4

**Summary:**

**Summary:**

The paper addresses the problem of *over-personalization* in LLM-based assistants and seeks a *rational equilibrium*, conceptually a Pareto-optimal balance, between personalization and generalization. It frames personalization as a **multi-objective reasoning task**, proposing the **Rational Personalization Acts (RPA)** framework, the **RPEVAL** benchmark, and the **RP-Reasoner** model, which performs pragmatic inference to decide *when and how* to use memory.

**Strengths:**

### **Strengths**

* **Timely and Well-Motivated Problem Setting.**
  The paper addresses an important and underexplored challenge in LLM-based personalized assistants—how to balance personalization and generalization by reasoning about *when and how* to apply user memory. The framing of personalization as a multi-objective pragmatic reasoning problem is both novel and relevant to current trends in LLM alignment.

* **High-Quality Benchmark (RPEVAL).**
  The **RPEVAL** This dataset can serve as a reusable diagnostic tool for evaluating rational memory utilization across future LLMs.

* **Effective and Practical Solution (RP-REASONER).**
  The proposed **RP-REASONER** model demonstrates large and consistent gains over strong baselines—improving intent prediction accuracy by roughly **35%** and reducing error severity by **26%**. Moreover, the finding that it resolves **≈80% of bad cases** in a real commercial assistant underscores its potential practical value and real-world applicability.

**Weaknesses:**

**Weaknesses:**

1. **Oversimplification of intent categories**
   The three-way scheme {Ignore, Support, Dominate} is a substantial simplification of real user needs. In practice, intentions are often multi-faceted, evolving, and context-dependent (e.g., conflicting or partially overlapping preferences).

2. **Unclear baseline motivation (Vanilla, Reminder, CoT)**
   The paper does not clearly justify why only Vanilla, Reminder, and CoT are used as baselines. It would strengthen the work to explain why **more advanced training-free approaches** and **stronger prompting ensembles** (e.g., self-consistency, self-refine/verify) are omitted. Without this rationale, it remains unclear whether the reported gains derive from the proposed pragmatic reasoning itself or from a limited baseline set.

3. **Conceptual overlap with prior work**
   The problem formulation is closely related to [1], which likewise argues that LLMs should not naively trust historical personalization and must continually detect and adapt to shifting user preferences. Both are **training-free, inference-time** frameworks that dynamically correct misalignment between user preferences and model behavior. The paper should explicitly compare and position its contribution relative to [1].

Reference
[1] Unlearning Misalignment for Personalized LLM Adaptation via Instance-Response-Dependent Discrepancies (TMLR 2025).

**Questions:**

Question 1 (Baselines & Rationale)
Your baselines focus on prompting (Vanilla/Reminder/CoT). Why were advanced training-free approaches like [1] not included, and how would RP-Reasoner compare to stronger prompting ensembles (e.g., self-consistency [2], self-refine [3])?

Question 2 (Comparison & Positioning)
Rational Preference Utilization performs inference-time pragmatic reasoning in intent space to regulate memory usage. This is conceptually similar to [1], which performs training-free, inference-time discrepancy unlearning via probabilistic marginalization in response space.
Could you include a comparison with [1] and discuss where RP-Reasoner is preferable (e.g., interpretability, latency, robustness to stale or contradictory memories) and where [1] is stronger?

References

[1] Unlearning Misalignment for Personalized LLM Adaptation via Instance-Response-Dependent Discrepancies (TMLR 2025).

[2] Self-Consistency Improves Chain of Thought Reasoning in Language Models.

[3] Self-Refine: Iterative Refinement with Self-Feedback.

---

> ### Author Response · Authors · 2025-11-22
> **Rebuttal by Authors [W1, W2 & Q1]**
>
> **Response to Weakness 1: Oversimplification of intent categories**
>
> 1. We deliberately adopt a three-way annotation scheme to keep the label space sufficiently compact, thereby ensuring annotation quality and consistency. At the same time, this scheme is expressive enough to fully capture the applicability between a single atomic preference and the current query. On top of this, we expand to a 3^n **compositional intent space** via *single2multi* strategies. Thus, the model ultimately handles a combinatorial intent task rather than a simple three-class problem.
> 2. In practical memory systems, conflicting or overlapping preferences are typically resolved during the **memory management stage** (e.g., merging, updating, deduplication) before retrieval. RPEval focuses on evaluating how an LLM **utilizes retrieved memories** when generating responses. Under this task boundary, the current intent formulation sufficiently covers the key usage scenarios of real memory-augmented agents.
>
> ---
>
> **Response to Weakness 2 & Question 1: Concerns of baselines**
>
> Thanks for the suggestion！ We have added self-consistency [2] and self-refine [3] as additional baselines. Both methods incur reasoning costs that are equal to or higher than those of RP-Reasoner：
>
>
> | Method            | MACRO-IA | MACRO-LKN | MACRO-ALL | MICRO-IA | MICRO-LKN | MICRO-ALL |
> | :---------------- | -------- | --------- | --------- | -------- | --------- | --------- |
> | GPT-4.1           | 0.05     | 0.01      | 0.03      | 0.48     | 0.51      | 0.50      |
> | + CoT-SC [2]      | 0.22     | 0.08      | 0.13      | 0.56     | 0.61      | 0.59      |
> | + Self-refine [3] | 0.18     | 0.07      | 0.11      | 0.57     | 0.56      | 0.57      |
> | + RP-Reasoner     | **0.38** | **0.20**  | **0.27**  | **0.69** | **0.65**  | **0.63**  |
> | GPT-5             | 0.00     | 0.04      | 0.03      | 0.31     | 0.43      | 0.40      |
> | + CoT-SC [2]      | 0.17     | 0.09      | 0.12      | 0.45     | 0.55      | 0.52      |
> | + Self-refine [3] | 0.18     | 0.03      | 0.09      | 0.46     | 0.45      | 0.45      |
> | + RP-Reasoner     | **0.38** | **0.23**  | **0.30**  | **0.71** | **0.69**  | **0.70**  |

---

> ### Author Response · Authors · 2025-11-22
> **Rebuttal by Authors [W3 & Q2]**
>
> **Response to Weakness 3 & Question 2: Comparison & Positioning with CM [1]**
>
> Thank you for the helpful comment and for pointing out the connection to [1]！We will analyze the differences between our work and CM from six perspectives.
>
> 1. **Different goals.** RP-Reasoner is designed to *regulate whether and how to use memory*. In contrast, CM aims to “reduce misalignment between model outputs and an individual user’s preferred responses,” i.e., to **make the model *more likely to follow past preferences***, but it does not explicitly reason about *when* personalization is appropriate. Within our framework, CM is more naturally viewed as strengthening L1-level personalization.
>
> 2. **Open-ended vs. closed-set outputs.** RP-Reasoner naturally handles fully open-ended response spaces and diverse user preferences, while [1] mainly operates over a fixed label set (e.g., StackExchange subsite tags, CLINC150 intents) and cannot directly correct free-form generations outside that label space.
>
> 3. **Dependence on exact matches.** CM are most effective when the current intent exactly matches historically labeled preferences. RP-Reasoner, by contrast, does not rely on such exact matches: it can infer when and how to personalize from pragmatic cues in the current query and retrieved memories, even for previously unseen situations
>
> 4. **Empirical comparison.** Although CM does not develop a mechanism for *gating* memory use, its appendix suggests using semantic similarity to decide when preferences apply. Following this idea, we implement a CM-style baseline that uses a sentence-transformer to compute similarity between the current query and stored preferences, enabling preference-based correction only when similarity exceeds a threshold and otherwise ignoring the memory. On RPEval, we find that similarity alone is often insufficient to reliably decide when personalization is appropriate:
>
> 5. |   Method    | Ignore | Support | Dominate | ALL  |
>    | :---------: | ------ | ------- | -------- | ---- |
>    | Similarity  | 0.3    | 0.26    | 0.52     | 0.36 |
>    | RP-Reasoner | 0.7    | 0.7     | 0.9      | 0.77 |
>
>    **Complementarity.** Conceptually, the two approaches can be combined: when a user has expressed a stable query–preference pattern in the past, CM can be applied on top of the base model’s prediction; in more general or open-ended scenarios without an exact historical match, RP-Reasoner can directly infer from the current query and memories how to perform rational, preference-aware generation.
>
> 6. **Timeline.** Finally, we note that [1] is nearly concurrent with our work: it was released in September 2025, and our submission was made in the same month, so the two lines of work should be viewed as independent, parallel explorations.
>
>
>
> References
>
> [1] Unlearning Misalignment for Personalized LLM Adaptation via Instance-Response-Dependent Discrepancies (TMLR 2025).
>
> [2] Self-Consistency Improves Chain of Thought Reasoning in Language Models.
>
> [3] Self-Refine: Iterative Refinement with Self-Feedback.

---

> ### Author Response · Authors · 2025-11-28
>
> Dear Reviewer,
>
> As the author-reviewer discussion period will end soon, we would appreciate it if you could check our response to your review comments. This way, if you have further questions and comments, we can still reply before the author-reviewer discussion period ends. Thank you very much for your time!

---

### Author Response · Authors · 2025-12-02

We are thankful for the reviewers' thoughtful feedback and insightful comments. Below, we provide a concise summary of the rebuttal:

We propose RPA , a framework with a benchmark and method to analyze the dual effects of memory and enable rational personalization.

All three reviewers acknowledged multiple strengths of the paper:
- **Important and timely problem setting.** Reviewers `vGZm`, `tTCm`, and `Gvb6` agree that over-personalization in LLM assistants is a real and underexplored issue, and that finding a rational balance between personalization and generalization is an important problem. Reviewer `tTCm` specifically highlighted the sleep-music example in Figure 1 as an intuitive illustration.
- **Value of the RPEVAL benchmark.** Reviewers `vGZm` and `tTCm` found our dataset construction strategy to be clever, and noted that the error taxonomy is useful for systematically understanding failure modes . Reviewer `tTCm` also emphasized the counterintuitive finding that larger models can perform worse at ignoring irrelevant preferences on RPEVAL, which is itself informative for the community.
- **Effective and practical solution.** Reviewers `vGZm` and `tTCm` praised RP-Reasoner for its strong empirical performance and practical relevance.

---
On top of these shared strengths, the reviewers raised several important concerns:

- **Position**
  - Reviewer `vGZm`  asked for a clearer comparison between RP-Reasoner and CM.
  - Reviewer `Gvb6` felt that our core idea is close to traditional context-aware recommendation.
  - Reviewer `Gvb6` pointed out that our use of the terms *memory utilization* and *risk* is narrower than what they expected.

- **Data quality and evaluation protocol**
  - Reviewer `vGZm` pointed that our intent space simplifies real user needs.
  - Reviewer `tTCm` raised concerns about a possible “closed loop” and asked for details on inter-annotator agreement and disagreement resolution.
  - Reviewer `Gvb6` questioned the reliability of the LLM-as-judge setup, and challenged the “filter bubble” example in the appendix.

- **Baselines and cost analysis**
  - **Baseline choice and implementation.** Reviewer `vGZm` asked for comparisons with stronger  baselines. Reviewer `Gvb6` requested more transparency about how the baselines are implemented.
  - **Inference cost and Real experiments.**  Reviewer tTCm asked us to report the inference cost and latency, and to describe the real experimental setup.

- **Insufficient methodological analysis**

  - Reviewer `tTCm` felt that our analysis of the gap between humans and models is not detailed enough, and requested a more principled explanation.


During the discussion period, we carefully addressed each of these points and revised the manuscript accordingly. The main changes are summarized below:

- **Positioning and related work**
  - Provided a systematic comparison between RP-Reasoner and CM, and clearly distinguish our setting from context-aware recommendation in terms of problem formulation, data, and methodology.
  - Clarified the scope of *memory utilization* and *risk* in our paper.

- **Data quality and evaluation protocol**
  - Explained the benefits of using a compact intent space and its coverage after combinatorial extension.
  - Provided carefully human-designed data generation guidelines, which help mitigate the potential data generation–evaluation closed-loop issue.
  - Reported inter-annotator agreement statistics, and describe the full disagreement resolution and adjudication process.
  - Reported the agreement metrics  used to validate the LLM-as-judge, and clarify the definitions of the error types .

- **Baselines and Cost**
  - Added two stronger reasoning-based baselines: self-consistency and self-refine.
  - Added an inference cost analysis and provide more complete baseline implementations.

- **Result analysis**
  - Added fine-grained error analyses to decompose the remaining gap between humans and models into specific error types.
  - Added an interpretability-oriented experiment showing that the ubiquitous attraction bias in LLMs is a key reason why they struggle to truly discard existing context.

Before the software issue on November 27, the rating updates were as follows:

- **(Nov 22, 2025, 17:08 AoE) Reviewer** **Gvb6** **raised the score from 2 to 4**, acknowledging our clarifications and efforts on positioning and baseline design, while still sharing some of the other reviewers’ concerns about data and evaluation. We hope our detailed responses to other reviewers will resolve this issue.

- Reviewer **vGZm** and **tTCm** might have been occupied during the discussionwas unable to engage in discussion.

Below is the rating summary before and after rebuttal:

|Reviewer|vGZm|tTCm|Gvb6|
|--|--|--|--|
|Initial Rating |6|4| 2|
|Updated Rating |6|4| 4|

We sincerely thank the reviewers and ACs for their constructive feedback and guidance. Hope this summary will assist in making the final decision.

---

### Note · Authors · 2026-01-04

I have read and agree with the venue's withdrawal policy on behalf of myself and my co-authors.